# Diverse Text-to-Image Generation via Contrastive Noise Optimization

**Byungjun Kim**[*1]    **Soobin Um**[*2,1]    **Jong Chul Ye**[†1]

[1] Graduate School of AI, KAIST, Daejeon 34141, Republic of Korea
[2] Department of AI, Kookmin University, Seoul 02707, Republic of Korea
{app54781,jong.ye}@kaist.ac.kr   soobin.um@kookmin.ac.kr

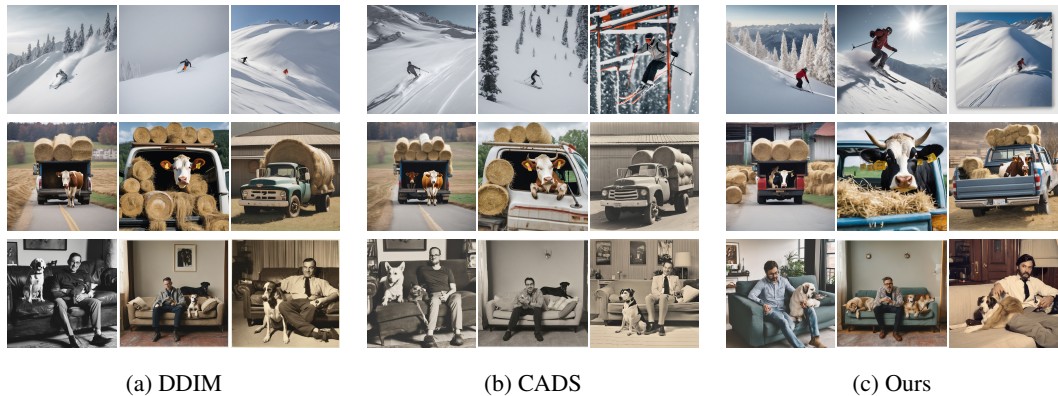

(a) DDIM                    (b) CADS                    (c) Ours

Figure 1: **Example results from our diverse image generation approach.** Three distinct prompts are used: (top) "A person skiing on a very snowy slope", (middle) "A cow sits in a truck with hay barrels in it", and (bottom) "A man sitting on a couch next to a dog". Standard DDIM (a) exhibits pronounced mode collapse, producing repetitive images and often failing to capture complex compositional details. CADS (Sadat et al., 2024) (b) improves diversity but still yields limited variation and occasional prompt misalignment. Our method (c) delivers markedly greater diversity and fidelity, generating a wide range of images that remain strongly aligned with the input text.

## ABSTRACT

Text-to-image (T2I) diffusion models have demonstrated impressive performance in generating high-fidelity images, largely enabled by text-guided inference. However, this advantage often comes with a critical drawback: limited diversity, as outputs tend to collapse into similar modes under strong text guidance. Existing approaches typically optimize intermediate latents or text conditions during inference, but these methods deliver only modest gains or remain sensitive to hyperparameter tuning. In this work, we introduce Contrastive Noise Optimization, a simple yet effective method that addresses the diversity issue from a distinct perspective. Unlike prior techniques that adapt intermediate latents, our approach shapes the initial noise to promote diverse outputs. Specifically, we develop a contrastive loss defined in the Tweedie data space and optimize a batch of noise latents. Our contrastive optimization repels instances within the batch to maximize diversity while keeping them anchored to a reference sample to preserve fidelity. We further provide theoretical insights into the mechanism of this preprocessing to substantiate its effectiveness. Extensive experiments across multiple T2I backbones demonstrate that our approach achieves a superior quality-diversity Pareto frontier while remaining robust to hyperparameter choices.

---

[*]These authors contributed equally to this work.
[†]Corresponding author.

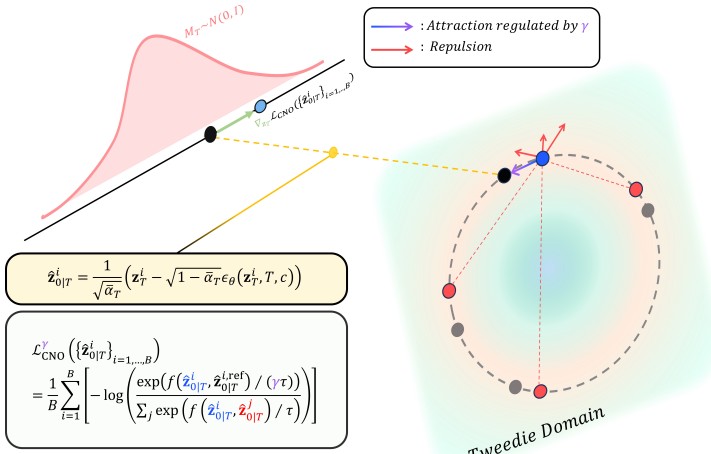

Figure 2: **Conceptual overview of contrastive noise optimization.** Our method enhances generation diversity by optimizing the initial latent vectors, $\mathbf{z}_T$, prior to the DDIM sampling process. We employ an InfoNCE loss that operates on a batch of noise vectors. This loss function pushes the optimizing sample (blue dot) away from all other negative samples in the batch to maximize separation. To preserve semantic fidelity, this repulsion is counterbalanced by an attraction force that pulls the anchor towards its original, non-optimized version (the positive pair), which acts as a fixed reference point. The attraction coefficient $\gamma$ regulates this anchoring force, stabilizing the fidelity-diversity trade-off. This pre-processing step effectively diversifies the final image outputs without fine-tuning or altering the foundational diffusion sampler.

# 1 INTRODUCTION

In recent years, diffusion models (Ho et al., 2020; Song et al., 2021; Rombach et al., 2022) have emerged as the leading paradigm for text-to-image (T2I) generation. A key driver of their success is the use of text-guided inference, which steers the generation process to produce images that are not only high-fidelity but also closely aligned with a given prompt. To maximize this alignment and enhance image quality, practitioners often employ strong guidance mechanisms, with techniques like Classifier-Free Guidance (CFG) (Ho & Salimans, 2021) becoming a standard practice. However, this pursuit of high fidelity comes at a significant cost: a pronounced lack of diversity. Under strong textual guidance, the model's outputs often collapse into a few dominant modes, failing to capture the rich variety of interpretations a text prompt can have. This fidelity-diversity trade-off (Dhariwal & Nichol, 2021) remains a critical bottleneck, severely restricting the creative potential of T2I models.

To address this challenge, a common line of work has focused on interventions during the iterative denoising process. These approaches typically optimize intermediate latents (Corso et al., 2024; Kirchhof et al., 2025) or manipulate text embeddings (Sadat et al., 2024; Um & Ye, 2025b) to enforce separation between samples, while other strategies rely on multi-agent systems or complex fine-tuning schedules (Ghosh et al., 2017). Although these methods have shown promise, they often require repeated adjustments during sampling and thus suffer from notable limitations, including substantial computational overhead (*e.g.*, DiversityPrompt (Um & Ye, 2025b); see Table 8) and limited gains under accelerated few-step models, where opportunities for intervention are intrinsically scarce.

In this work, we tackle the diversity problem at its fundamental source by shifting the paradigm from inference-time interventions to initial-noise selection. We introduce **Contrastive Noise Optimization (CNO)**, a simple yet powerful one-shot preprocessing framework that optimizes a batch of initial noise latents before sampling begins. Unlike prior methods that intervene repeatedly during the denoising trajectory, CNO performs a single, lightweight optimization step on initial noise and requires no modifications or adjustments during sampling. The key idea is to incorporate a

contrastive objective to shape a diversity-encouraging initial noise distribution, drawing inspiration from the structure of InfoNCE-based contrastive learning (van den Oord et al., 2019). Building on this foundation, we introduce structural modifications tailored to the unique demands of diverse T2I generation, enabling diversity to be enhanced while retaining strong semantic fidelity.

At the core of our framework lies a balance between two complementary forces: an attraction term that anchors each optimized noise to its original counterpart, and a repulsion term that encourages semantic separation across samples. Crucially, we impose this contrastive structure not in the raw noise latent space but in the Tweedie denoised prediction space, which provides the diffusion model's best estimate of clean data. Operating in this space allows the optimization to act directly on meaningful semantic signals, making the refinement of initial noise substantially more effective.

To further control this interplay, we introduce a novel balancing parameter $\gamma$, which modulates the relative strength of attraction and repulsion. Its behavior is analytically grounded through an extended mutual-information perspective (Proposition 2), revealing how $\gamma$ governs the contributions of positive and negative pairs. We additionally provide a closed-form selection rule for $\gamma$ that ensures robust behavior across batch sizes. Practical components such as adaptive latent pooling and a stop-gradient mechanism further enhance the efficiency and scalability of CNO for modern T2I backbones.

Comprehensive experiments demonstrate that our lightweight preprocessing substantially improves diversity across modern T2I frameworks (including Stable Diffusion (Rombach et al., 2022)) while maintaining high image quality and text alignment. Notably, CNO delivers consistent improvements even under accelerated few-step samplers such as FLUX (Labs et al., 2025) and SDXL-Lightning (Lin et al., 2024), where existing diversity approaches exhibit limited gains (see Section D.1).

- We introduce a **paradigm shift** from inference-time interventions to *initial-noise selection*, addressing the diversity problem at its source while eliminating repeated per-step adjustments and remaining effective even in accelerated few-step samplers.

- We develop a **contrastive noise optimization framework** tailored for diverse T2I generation, featuring the balancing parameter $\gamma$, an extended mutual-information analysis, and efficient heuristics such as adaptive latent pooling and stop-gradient.

- We demonstrate **state-of-the-art diversity–quality trade-offs** across major T2I backbones, including SD1.5, SDXL, SD3, and fast few-step samplers such as FLUX and SDXL-Lightning.

## 2 RELATED WORK

Improving the diversity of diffusion models has recently attracted much attention, mainly due to their increasing use in critical applications such as text-to-image generation (Rombach et al., 2022). One prominent effort is CADS (Sadat et al., 2024), which enhances sample diversity by gradually annealing noise perturbations on conditional embeddings. Although effective, their approach is sensitive to the noise annealing schedule and requires laborious hyperparameter searches to see the diversity gain. A fundamentally different strategy is seen in Particle Guidance (PG) (Corso et al., 2024). The idea is to repel intermediate latent samples that share the same condition, thereby encouraging the final generated samples to exhibit distinct features. While it does not require difficult parameter searches like CADS, it often provides limited diversity gains (Kirchhof et al., 2025). This approach shares a similar spirit as PG (Corso et al., 2024) and incorporates diversity-improving guidance for repelling intermediate latent instances during inference, yet in a sparse manner, *i.e.*, not at every inference timestep. A key distinction from ours is that its diversity optimization (by injecting guidance) is performed over inference time, which may be more expensive compared to ours that focuses on the initial latent space.

A related yet different task is to generate *minority* samples – low-density instances in the data manifold (Sehwag et al., 2022; Um & Ye, 2023; 2025b). Pioneer works in this area are offered by Sehwag et al. (2022); Um & Ye (2023), which share a similar idea of incorporating classifier guidance (Dhariwal & Nichol, 2021) to push intermediate samples toward low-density regions. The reliance on external classifiers was addressed in Um & Ye (2024; 2025b); Um et al. (2025), offering self-contained approaches for producing minority samples with diffusion models. While relevant,

the task of generating low-density minority samples is distinct from improving diversity and does not guarantee distinct outputs. Notably, MinorityPrompt (Um & Ye, 2025b) considers text-to-image generation and provides a prompt optimization framework that can also be used to enhance the diversity of generated samples. However, it requires optimizing the diversity-improving prompt during inference, which imposes substantial computational overhead (Um & Ye, 2025b).

Initial noise optimization in diffusion models has been explored in various contexts (Guo et al., 2024; Ahn et al., 2024). One instance is InitNO (Guo et al., 2024), where the idea is to optimize the initial noise latent to promote improved prompt alignment in text-to-image generation. A distinction with respect to ours is that their focus is on enhancing text adherence, unlike ours. Another notable work was done by Ahn et al. (2024), who aim to characterize the influence of classifier-free guidance (Ho & Salimans, 2021) through a properly optimized latent noise, enabled by an additional neural network that maps to the optimal noise. While interesting, their focus is inherently distinct from ours. To the best of our knowledge, our framework is the first to incorporate the idea of noise optimization for addressing the diversity challenge of diffusion models.

## 3 PRELIMINARIES

### 3.1 LATENT DIFFUSION MODELS

Latent Diffusion Models (LDMs) (Rombach et al., 2022) improve upon traditional Denoising Diffusion Probabilistic Models (DDPMs) (Ho et al., 2020) by performing the diffusion process in a computationally efficient, lower-dimensional latent space. LDMs first use a pre-trained autoencoder to map a high-resolution image $\mathbf{x}_0$ into a compressed latent representation, $\mathbf{z}_0 = \mathcal{E}(\mathbf{x}_0)$. The diffusion process is then applied directly to these latent vectors.

The forward process is a Markov chain that gradually adds Gaussian noise to an initial latent vector $\mathbf{z}_0$ over a series of $T$ discrete timesteps. At each step $t$, the transition is defined as:

$$q(\mathbf{z}_t|\mathbf{z}_{t-1}) = \mathcal{N}(\mathbf{z}_t; \sqrt{1-\beta_t}\mathbf{z}_{t-1}, \beta_t\mathbf{I}), \tag{1}$$

where $\{\beta_t\}_{t=1}^T$ is a fixed variance schedule that controls the noise level at each step. A key property of this process is that the marginal distribution at any arbitrary step $t$ can be expressed in a closed form conditioned only on the initial latent $\mathbf{z}_0$:

$$q(\mathbf{z}_t|\mathbf{z}_0) = \mathcal{N}(\mathbf{z}_t; \sqrt{\bar{\alpha}_t}\mathbf{z}_0, (1-\bar{\alpha}_t)\mathbf{I}), \tag{2}$$

where we define $\alpha_t := 1 - \beta_t$ and $\bar{\alpha}_t := \prod_{i=1}^t \alpha_i$. As $t$ increases towards $T$, the signal term $\sqrt{\bar{\alpha}_t}$ approaches zero, and the variance $1 - \bar{\alpha}_t$ approaches 1. This ensures that the noised latent $\mathbf{z}_T$ reliably converges to an isotropic Gaussian distribution $\mathcal{N}(\mathbf{0}, \mathbf{I})$, regardless of the initial latent vector $\mathbf{z}_0$. Once the reverse process generates a clean latent, the decoder $\mathcal{D}$ is used to map it back to the pixel space.

### 3.2 REVERSE PROCESS AND DENOISING VIA TWEEDIE'S FORMULA

The generative process is achieved by reversing the forward process, conditioned on external information such as a text embedding $\mathbf{c}$ for Text-to-Image (T2I) synthesis. This involves learning a model $p_\theta(\mathbf{z}_{t-1}|\mathbf{z}_t, \mathbf{c})$ that approximates the true posterior. In DDPM (Ho et al., 2020), this conditional reverse process is parameterized as a Gaussian whose mean is learned by a neural network $\boldsymbol{\epsilon}_\theta(\mathbf{z}_t, t, \mathbf{c})$:

$$p_\theta(\mathbf{z}_{t-1}|\mathbf{z}_t, \mathbf{c}) = \mathcal{N}\left(\mathbf{z}_{t-1}; \frac{1}{\sqrt{\alpha_t}}\left(\mathbf{z}_t - \frac{\beta_t}{\sqrt{1-\bar{\alpha}_t}}\boldsymbol{\epsilon}_\theta(\mathbf{z}_t, t, \mathbf{c})\right), \sigma_t^2\mathbf{I}\right). \tag{3}$$

The core of this process is the network $\boldsymbol{\epsilon}_\theta$, which is trained to predict the noise component from the noisy latent vector $\mathbf{z}_t$ based on the condition $\mathbf{c}$. The key insight is that this trained network can be used to directly estimate the original clean latent $\mathbf{z}_0$ at any timestep $t$. This denoised estimate $\hat{\mathbf{z}}_0$ is implemented via Tweedie's formula (Chung et al., 2025; Um & Ye, 2025a), which for our specific noise model takes the form:

$$\hat{\mathbf{z}}_{0|t}(\mathbf{z}_t, t, \mathbf{c}) := \frac{1}{\sqrt{\bar{\alpha}_t}}\left(\mathbf{z}_t - \sqrt{1-\bar{\alpha}_t}\boldsymbol{\epsilon}_\theta(\mathbf{z}_t, t, \mathbf{c})\right). \tag{4}$$

This equation forms the foundation of the iterative denoising process in many conditional diffusion models, allowing for the generation of latent vectors that align with the given context $\mathbf{c}$.

### 3.3 Information Noise-Contrastive Estimation (InfoNCE)

Information Noise-Contrastive Estimation (InfoNCE) (van den Oord et al., 2019) is a fundamental objective for self-supervised representation learning (Chen et al., 2020; Kim et al., 2021; Jang et al., 2023; Kim et al., 2025). It aims to construct an embedding space that maximizes mutual information (Cover, 1999) between representations of positive (similar) pairs while minimizing it for negative (dissimilar) pairs. The learning process can be viewed as a classification task in which, for a given anchor sample, the model must correctly identify its positive counterpart from a set of negative samples.

Specifically, for an anchor embedding vector $\mathbf{z}_i$, its positive pair $\mathbf{z}_i^{\text{pos}}$, and a set of $B - 1$ negative samples $\{\mathbf{z}_j\}_{j=1, j\neq i}^{B}$, the InfoNCE loss is formulated as

$$\mathcal{L}_{\text{InfoNCE}} := \frac{1}{B} \sum_{i=1}^{B} \left[ -\log \left( \frac{\exp(f(\mathbf{z}_i, \mathbf{z}_i^{\text{pos}})/\tau)}{\sum_{j=1}^{B} \exp(f(\mathbf{z}_i, \mathbf{z}_j)/\tau)} \right) \right], \tag{5}$$

where $f(\cdot, \cdot)$ denotes a similarity measure (*e.g.*, cosine similarity) between two representation vectors, and $\tau$ is a temperature parameter controlling the sharpness of the distribution. Intuitively, the loss encourages the anchor to be close to its positive pair while pushing it away from all negative samples, thereby tightening intra-class similarity and enlarging inter-class separation in the embedding space.

## 4 Proposed Method

### 4.1 Optimizing Initial Noise with Contrastive Loss between Tweedies

A core challenge in text-to-image diffusion models is that independently sampled initial noises $\mathbf{z}_T$ often lead to generations that collapse into similar modes, even under varied stochasticity. Rather than intervening during the denoising trajectory as in prior diversity methods, we enhance diversity at its fundamental source by *optimizing the initial noises themselves* before sampling begins. Our approach, which we call Contrastive Noise Optimization (CNO), refines a batch of initial noises using a contrastive objective applied in the Tweedie denoised prediction space, enabling the noises to be semantically well-separated while remaining faithful to their original distribution. The full procedure is provided in Algorithm 1 in Section B.1.

The algorithm proceeds as follows. First, we sample a batch of initial latent codes $\mathbf{Z}_T = \{\mathbf{z}_T^i\}_{i=1}^{B}$ from a standard Gaussian distribution $\mathcal{N}(0, \mathbf{I})$. Hereafter, we denote the denoised estimate from Equation Eq. (4) as $\hat{\mathbf{z}}_{0|t}$. Using this, we compute the initial target latents, $\{\hat{\mathbf{z}}_{0|T}^i\}_{i=1}^{B}$, by applying the denoising estimator defined in Equation Eq. (4) to this initial noise at timestep $T$. Each resulting $\hat{\mathbf{z}}_{0|T}^i$ is therefore the model's one-step prediction of the clean latent $\mathbf{z}_0$ from the noise $\mathbf{z}_T^i$. These pre-computed latents then serve as fixed anchors, each defining a unique identity for its respective sample throughout the optimization.

Before computing the loss, we employ a practical optimization to enhance efficiency. The high dimensionality of the latents $(B, C, S, S)$ makes the pairwise similarity calculation computationally intensive. We found experimentally that applying an adaptive average pooling operation to downsample the latents to a sufficiently smaller spatial resolution $(B, C, w, w)$, where $w < S$, did not compromise performance (Section 5.1). This step substantially reduces memory usage and accelerates the similarity matrix computation, making the optimization process more tractable. Downsampled latents $\{\hat{\mathbf{z}}_{0|T}^{i,\text{ref}}\}_{i=1}^{B}$, $\{\hat{\mathbf{z}}_{0|T}^i\}_{i=1}^{B}$ are then normalized, and the noise $\{\mathbf{z}_T^i\}_{i=1}^{B}$ is updated using a contrastive loss $\mathcal{L}_{\text{CNO}}$:

$$\mathcal{L}_{\text{CNO}} := \frac{1}{B} \sum_{i=1}^{B} \left[ -\log \left( \frac{\exp(f(\hat{\mathbf{z}}_{0|T}^i, \hat{\mathbf{z}}_{0|T}^{i,\text{ref}})/\tau)}{\sum_{j=1}^{B} \exp(f(\hat{\mathbf{z}}_{0|T}^i, \hat{\mathbf{z}}_{0|T}^j)/\tau)} \right) \right]. \tag{6}$$

This loss function is designed to achieve two objectives simultaneously.

**Attraction (Numerator).** It encourages the current latent $\hat{\mathbf{z}}_{0|T}^i$ to remain similar to its corresponding initial target latent $\hat{\mathbf{z}}_{0|T}^{i,\text{ref}}$. This ensures that each sample maintains coherence with its initial concept and does not drift away during optimization.

**Repulsion (Denominator).** It pushes the current latent $\hat{\mathbf{z}}_{0|T}^i$ to be dissimilar from all other current latents $\{\hat{\mathbf{z}}_{0|T}^j\}_{j=1, j \neq i}^B$ in the batch. This directly promotes diversity by forcing the latent representations to disperse within the batch.

By iteratively updating $\mathbf{z}_T$ with the gradient of this loss ($\nabla_{\mathbf{z}_T} \mathcal{L}_{\text{CNO}}$), we guide the initial noise vectors to positions in the latent space that are predisposed to generating a diverse set of images. Once the optimization is complete, this well-distributed batch of noise $\{\mathbf{z}_T^i\}_{i=1}^B$ is fed into a standard, pre-trained DDIM denoiser to produce the final images. Consequently, our method effectively enhances output diversity through a simple pre-processing stage that modulates the starting point of the generation, all without requiring any modifications to the pre-trained diffusion model itself.

**Stop-gradient for computational efficiency.** Optimizing the loss in Eq. (6) requires backpropagation through diffusion models, which can incur substantial computational overhead. To mitigate this, we apply a `stopgrad` operator (Chen & He, 2021) on the model path used in computing the Tweedie's estimate. As also demonstrated in Ahn et al. (2024), this simple strategy yields significant savings in training cost with only marginal impact on performance (see Table 8).

### 4.2 Gamma Effect: Regulated Attraction for Stable Image Diversification

In our proposed algorithm, the InfoNCE loss for a single sample within a batch of size $B$ consists of one attraction term (to itself) and $B - 1$ repulsion terms (from all other samples in the batch). When the batch size $B$ is large, the cumulative repulsion force can become excessively strong. This risks pushing the optimized noise out of the intended distribution, potentially leading to the generation of less plausible or out-of-distribution images.

To mitigate this issue and achieve a more stable optimization, we introduce a coefficient, $\gamma$, to dynamically regulate the attraction force. This is done by dividing the similarity term in the numerator of the loss function by $\gamma$. The modified InfoNCE loss is as follows:

$$\mathcal{L}_{\text{CNO}}^\gamma := \frac{1}{B} \sum_{i=1}^B \left[ -\log \left( \frac{\exp(f(\hat{\mathbf{z}}_{0|T}^i, \hat{\mathbf{z}}_{0|T}^{i,\text{ref}})/(\gamma\tau))}{\sum_{j=1}^B \exp(f(\hat{\mathbf{z}}_{0|T}^i, \hat{\mathbf{z}}_{0|T}^j)/\tau)} \right) \right]. \tag{7}$$

Empirically, we found that $\gamma$ in our framework behaves similarly to a Gaussian regularizer (Guo et al., 2024), which penalizes large deviations from the Gaussian prior. A detailed analysis is provided in Section C.3.

**Desirable Value for $\gamma$.** The desirable value for $\gamma$ is derived by creating a balance between the regulated attraction force and the cumulative repulsion forces. We achieve this by equating the maximum value of the attraction term (numerator) with the sum of the maximum values of the $B - 1$ repulsion terms. Assuming the maximum similarity score is 1, this balance can be expressed as:

$$\exp(1/(\gamma\tau)) = (B-1)\exp(1/\tau). \tag{8}$$

Solving for $\gamma$ gives us the following relationship:

$$\gamma = (\tau ln(B-1) + 1)^{-1}. \tag{9}$$

For instance, in our common experimental setting where $\tau = 0.1$ and $B = 5$, the calculated $\gamma$ is approximately 0.88, which is very close to the fixed value of $\gamma = 1.0$ we have consistently used. For a fixed $\tau = 0.1$, the optimal $\gamma$ changes moderately with batch size $B$:

$$B = 13 \rightarrow \gamma \approx 0.8 \quad B = 73 \rightarrow \gamma \approx 0.7 \quad B = 775 \rightarrow \gamma \approx 0.6$$

This shows that as the batch size $B$ grows larger, $\gamma$ is not highly sensitive. Therefore, using a single, appropriately chosen fixed value for $\gamma$ can also yield stable results without significant performance degradation.

### 4.3 Theoretical Intuitions

We provide mathematical insights into our contrastive framework by establishing its connection to mutual information. We begin with the classical view of InfoNCE as a variational lower bound

on mutual information, as shown by van den Oord et al. (2019). Specifically, the InfoNCE loss in Eq. (5) satisfies

$$\mathcal{L}_{\text{InfoNCE}} \geq \log B - I(Z; Z_{\text{pos}}), \tag{10}$$

where $I(X;Y)$ is the mutual information between random variables $X$ and $Y$. This inequality implies that minimizing $\mathcal{L}_{\text{InfoNCE}}$ indirectly maximizes $I(Z; Z_{\text{pos}})$, encouraging the learned embedding space to cluster positive pairs. However, this classical relationship does not clarify how negative pairs shape the embedding space – an aspect that is critical in our framework, where negative samples drive diversity.

To capture this effect, we augment the traditional bound to incorporate mutual information with respect to negative pairs. The following proposition formalizes this result.

**Proposition 1.** *The InfoNCE loss in Eq. (5) satisfies*

$$\mathcal{L}_{\text{InfoNCE}} \geq -I(Z; Z_{\text{pos}}) + I(Z; Z_{\text{neg}}) + \log(B - 1), \tag{11}$$

*where $B$ denotes the batch size, and $I(X;Y)$ is the mutual information between random variables $X$ and $Y$:*

$$I(X;Y) := \mathbb{E}_{p(X,Y)}\left[\log \frac{p(X,Y)}{p(X)p(Y)}\right] = \mathbb{E}_{p(X,Y)}\left[\log \frac{p(X \mid Y)}{p(X)}\right].$$

The proof is provided in Section A.1. This result shows that the InfoNCE loss is inherently linked to negative samples as well as positive ones: minimizing the loss decreases the mutual information with negatives while increasing that with positives.

**Extension with Gamma.** We further analyze our modified loss in Eq. (7), which introduces a coefficient $\gamma$ to control the relative strength of positive pairs.

**Proposition 2.** *For the loss function defined in Eq. (7), the following inequality holds:*

$$\mathcal{L}_{\text{CNO}}^{\gamma} \geq -\frac{1}{\gamma} I(Z; Z_{\text{pos}}) + I(Z; Z_{\text{neg}}) + \log(B - 1). \tag{12}$$

The proof is given in Section A.2. This proposition indicates that $\gamma$ scales the positive mutual information term, serving as a control knob to modulate the influence of positive pairs in our contrastive objective. We provide empirical results to demonstrate the impact of $\gamma$ in the appendix; see Section C.2.

## 5 EXPERIMENTS

**Implementation Details.** Our experiments are conducted on three distinct pre-trained text-to-image diffusion frameworks: Stable Diffusion v1.5 (SD1.5), SDXL, and SD3. We compare our method with state-of-the-art zero-shot diversity samplers, including Condition-Annealed Diffusion Sampler (CADS) (Sadat et al., 2024) and Particle Guidance (PG) (Corso et al., 2024), as well as the prompt-optimization-based diversity method of Um & Ye (2025b), referred to as *DiversityPrompt*. All evaluations use text prompts randomly sampled from the MS-COCO (Lin et al., 2014) validation set. For each prompt, we generate 3–5 images, yielding a total of roughly 6–10 K samples.

**Evaluation Metrics.** The goal of our research is to enhance the Pareto frontier between image quality and diversity of generated images while maintaining a high degree of relevance to the text prompt. To quantitatively assess this, we used the following key metrics: **CLIPScore, PickScore, Image-Reward** for evaluating image quality, and **Vendi Score, Mean Pairwise Similarity (MSS)** for diversity. Details for those metrics appear in Section B.2.

### 5.1 RESULTS

**Comparison with Existing Zero-Shot Diversity Samplers.** The results in Table 1 demonstrate the effectiveness of our approach. Our method achieves high performance on the key diversity metrics, Vendi Score and MSS, consistently outperforming all baselines across the different foundation models. While performance on Density and Coverage is highly competitive, our approach's strong results on Vendi Score and MSS prove its robust, model-agnostic ability to mitigate mode collapse.

| Model | Method | Prec ↑ | Rec ↑ | Den ↑ | Cov ↑ | CLIP ↑ | Pick ↑ | IR ↑ | MSS ↓ | Vendi ↑ |
|---|---|---|---|---|---|---|---|---|---|---|
| SD1.5 | DDIM | 0.7018 | 0.6706 | 0.6033 | 0.7382 | **31.5863** | **21.5081** | **0.2222** | 0.1657 | 4.6949 |
| | PG | 0.6940 | 0.7024 | 0.5975 | 0.7446 | 31.3222 | 21.2086 | 0.1712 | 0.1426 | 4.7630 |
| | CADS | 0.6866 | **0.7240** | 0.5686 | 0.7292 | 31.4863 | 21.2938 | 0.1137 | 0.1330 | 4.7805 |
| | DiversityPrompt | 0.6878 | 0.7006 | 0.5839 | 0.7416 | 31.5457 | 21.3510 | 0.1332 | 0.1393 | 4.7599 |
| | Ours | **0.7308** | 0.6926 | **0.6528** | **0.7728** | 31.4525 | 21.3779 | 0.1284 | **0.1317** | **4.7855** |
| SDXL | DDIM | **0.6858** | 0.6538 | **0.5713** | 0.7368 | 31.8788 | **22.4761** | **0.7302** | 0.2169 | 2.8377 |
| | PG | 0.5820 | **0.7088** | 0.3855 | 0.5606 | 31.5679 | 22.1631 | 0.6950 | 0.2050 | 2.8545 |
| | CADS | 0.6486 | 0.6796 | 0.5262 | 0.7108 | **31.9424** | 22.2078 | 0.6162 | 0.1765 | 2.8864 |
| | Ours | 0.6720 | 0.6992 | 0.5553 | **0.7568** | 31.8129 | 22.3859 | 0.7273 | **0.1623** | **2.9019** |
| SD3 | FM-ODE | 0.7184 | **0.5828** | 0.6472 | 0.6770 | 31.7783 | **22.5763** | 1.0301 | 0.3028 | 4.2205 |
| | PG | **0.7782** | 0.3900 | **0.8110** | **0.7370** | **32.0463** | 22.3500 | **1.0357** | 0.3066 | 4.2097 |
| | CADS | 0.6984 | 0.5752 | 0.6110 | 0.6682 | 31.6974 | 22.4987 | 1.0233 | 0.2960 | 4.2487 |
| | Ours | 0.7100 | 0.5806 | 0.6573 | 0.6938 | 31.7713 | 22.5647 | 1.0233 | **0.2909** | **4.2644** |

Table 1: **Quantitative results of zero-shot diverse samplers.** Our proposed method is benchmarked against standard samplers (DDIM and Flow-Matching ODE) and state-of-the-art diversity-enhancing techniques: PG (Corso et al., 2024) and CADS (Sadat et al., 2024). DiversityPrompt refers to the prompt-optimization-based diversity approach developed in Um & Ye (2025b). The evaluation demonstrates that our approach consistently achieves superior performance in diversity metrics, including MSS(↓) and Vendi Score(↑), across Stable Diffusion 1.5, XL, and 3. Notably, it enhances diversity while effectively preserving image quality and prompt fidelity, successfully navigating the fidelity-diversity trade-off by optimizing the initial latent space.

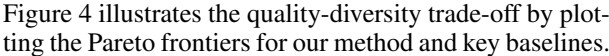

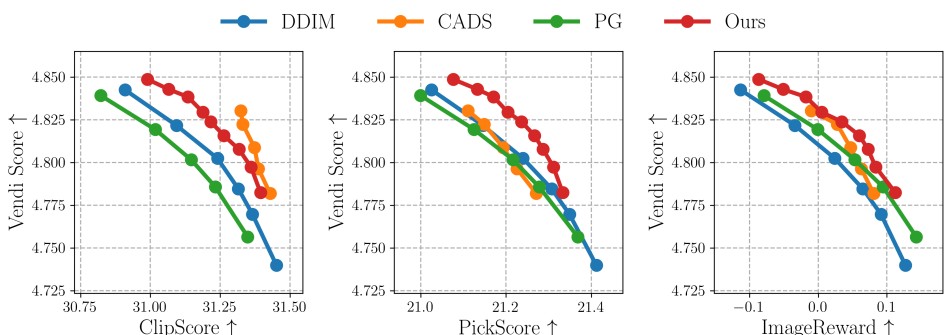

Figure 4: Pareto curves of diverse sampling methods between Vendi Score and text-to-image alignment metrics. For our methods, we use $N_{opt} = 5, \gamma = 1.0, w = 8, \tau = 0.1$ in common.

Crucially, these substantial gains in diversity do not compromise generation quality. Our method maintains strong prompt fidelity, evidenced by competitive CLIP scores, and sustains a competitive or superior Pick-Score compared to CADS across all Stable Diffusion models. This indicates our outputs are not only more varied but also aesthetically preferable. This quantitative strength is mirrored in our qualitative results (see Figure 5), where our model shows particular strength on complex compositional prompts that cause competitor methods to fail. Where gains in diversity often come at the cost of quality, our method achieves both, delivering outputs that are not only more varied but also consistently high in fidelity.

Figure 4 illustrates the quality-diversity trade-off by plotting the Pareto frontiers for our method and key baselines.

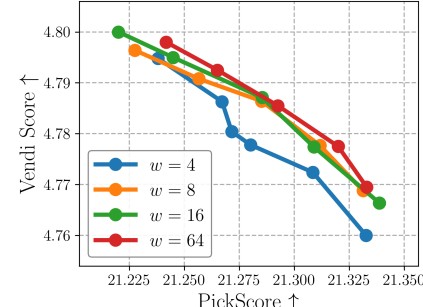

Figure 3: Ablation on the window size $w$. The Pareto frontier of PickScore vs. Vendi Score.

The plots reveal a clear and compelling advantage for our approach, which establishes a dominant frontier across metrics trained on large-scale human preferences. This is most evident in the **PickScore** and **Image-Reward** charts, where our method is strictly superior to competitors like CADS, indicating our outputs are more aesthetically pleasing for any given level of diversity. While

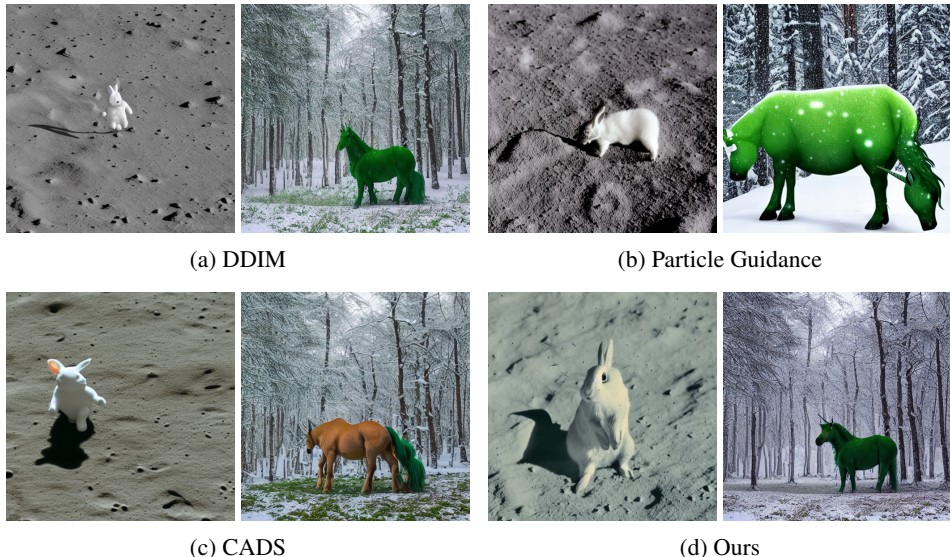

(a) DDIM  (b) Particle Guidance

(c) CADS  (d) Ours

Figure 5: **Qualitative comparison with pre-existing zero-shot diverse generative methods** For the prompt *"A white rabbit on the moon."*(left) and *"A green unicorn in a snowy forest"*(right), we compare our method (d) with baseline approaches. Our method successfully generates high-fidelity images that are strongly aligned with the text prompts. In contrast, the other methods exhibit various failures.

CADS may achieve a marginally higher peak CLIPScore, this metric is known to favor rigid semantic alignment rather than creative or aesthetically superior interpretations. In contrast, our method's dominance across both PickScore and Image-Reward demonstrates a more intelligent trade-off. It prioritizes what a human user would find visually appealing and contextually appropriate over a mechanical, word-for-word adherence to the prompt. This quantitative strength is mirrored in our qualitative results (see Appendix D.4), where our model uniquely succeeds on complex compositional prompts that cause competitors to fail.

**Effect of Window Size $w$.** We conduct an ablation study to investigate the effect of the downsampling window size, $w$, applied to the Tweedie latent shape $\hat{\mathbf{z}}_{0|T}^i$(Algorithm 1, Line 10). This step is crucial for capturing the global structure of the initial noise prediction while reducing computational cost. We experiment with $w \in \{4, 8, 16\}$ and compare these against the baseline of $w = 64$, which effectively uses the full-resolution latent shape.

The results are illustrated in the PickScore-Vendi Score Pareto frontier in Figure 3. As shown, an aggressive downsampling with $w = 4$ leads to a noticeable performance degradation, failing to match the frontier established by larger window sizes. In contrast, moderate downsampling with $w = 8$ **and $w = 16$ achieves highly competitive performance compared to the $w = 64$ baseline**. This suggests that moderate downsampling successfully preserves the essential structural information for diversification while benefiting from increased computational efficiency. Excessive downsampling ($w = 4$), however, appears to discard critical details necessary for the optimization process. Based on these findings, we select $w = 16$ for our main experiments, as it provides the best trade-off between performance and efficiency.

## 6 CONCLUSION

We introduced **Contrastive Noise Optimization**, a simple yet effective pre-processing method to address mode collapse in text-to-image (T2I) diffusion models. By applying a contrastive loss directly to the initial noise vectors for a given text prompt, our approach ensures diverse starting points for generation, eliminating the need for the complex sampling guidance or laborious hyperparameter tuning required by prior work. Our method sets a new state-of-the-art on the quality-diversity Pareto frontier, outperforming strong baselines on key diversity metrics without compromising prompt fidelity or image quality.

ACKNOWLEDGMENTS

This research was supported by the National Research Foundation of Korea (NRF) under Grant RS-2024-00336454, and the AI Computing Infrastructure Enhancement (GPU Rental Support) User Support Program funded by the Ministry of Science and ICT (MSIT), Republic of Korea (RQT-25-120217). This work was also supported by the Institute of Information & Communications Technology Planning & Evaluation (IITP) grant funded by the Korean government (MSIT) (No. RS-2019-II190075, Artificial Intelligence Graduate School Program(KAIST); No. RS-2025-02304967, AI Star Fellowship (KAIST); No. RS-2025-02219317, AI Star Fellowship (Kookmin University)).

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

# A  PROOFS

## A.1  PROOF OF PROPOSITION 1

**Proposition 1.** *The InfoNCE loss in Eq. (5) satisfies*

$$\mathcal{L}_{\text{InfoNCE}} \geq -I(Z; Z_{\text{pos}}) + I(Z; Z_{\text{neg}}) + \log(B-1), \tag{13}$$

*where $B$ denotes the batch size, and $I(X; Y)$ is the mutual information between random variables $X$ and $Y$:*

$$I(X; Y) := \mathbb{E}_{p(X,Y)} \left[ \log \frac{p(X,Y)}{p(X)p(Y)} \right] = \mathbb{E}_{p(X,Y)} \left[ \log \frac{p(X \mid Y)}{p(X)} \right].$$

*Proof.* In general, we can formulate the infoNCE loss as Eq. (14) by setting $g(\cdot, \cdot) = \exp(f(\cdot, \cdot)/\tau)$ from Eq. (5):

$$\mathcal{L}_{\text{InfoNCE}} = -\mathbb{E}_{p(\mathbf{z}, \mathbf{z}_{\text{pos}}, \mathbf{z}_{\text{neg}})} \left[ \log \frac{g(\mathbf{z}, \mathbf{z}_{\text{pos}})}{g(\mathbf{z}, \mathbf{z}_{\text{pos}}) + \sum_{i=1}^{B-1} g(\mathbf{z}, \mathbf{z}_{\text{neg}}^{(i)})} \right]. \tag{14}$$

Let us simply notate $\mathbf{z}_{\text{pos}}, \mathbf{z}_{\text{neg}}$ as $\mathbf{z}_p, \mathbf{z}_n$, respectively. This can be split into two terms:

$$\mathcal{L}_{\text{InfoNCE}} = -\mathbb{E}_{p(\mathbf{z}, \mathbf{z}_p)} \left[ \log g(\mathbf{z}, \mathbf{z}_p) \right] + \mathbb{E}_{p(\mathbf{z}, \mathbf{z}_p, \mathbf{z}_n)} \left[ \log \left\{ g(\mathbf{z}, \mathbf{z}_p) + \sum_{i=1}^{B-1} g(\mathbf{z}, \mathbf{z}_n^{(i)}) \right\} \right]. \tag{15}$$

Given that $g(\cdot, \cdot)$ is non-negative due to its exponential form, Eq. (15) has a lower bound by omitting the positive pair similarity from second term:

$$\mathcal{L}_{\text{InfoNCE}} \geq -\mathbb{E}_{p(\mathbf{z}, \mathbf{z}_p)} \left[ \log g(\mathbf{z}, \mathbf{z}_p) \right] + \mathbb{E}_{p(\mathbf{z}, \mathbf{z}_n)} \left[ \log \sum_{i=1}^{B-1} g(\mathbf{z}, \mathbf{z}_n^{(i)}) \right]. \tag{16}$$

According to van den Oord et al. (2019), function $g(\mathbf{z}, \mathbf{z}')$ estimates the probability density ratio $\frac{p(\mathbf{z} \mid \mathbf{z}')}{p(\mathbf{z})}$ related to mutual information maximization. This formulation of $g$ induces the following Eq. (17):

$$\mathcal{L}_{\text{InfoNCE}} \geq -\mathbb{E}_{p(\mathbf{z}, \mathbf{z}_p)} \left[ \log \frac{p(\mathbf{z} \mid \mathbf{z}_p)}{p(\mathbf{z})} \right] + \mathbb{E}_{p(\mathbf{z}, \mathbf{z}_n)} \left[ \log \sum_{i=1}^{B-1} \frac{p(\mathbf{z} \mid \mathbf{z}_n^{(i)})}{p(\mathbf{z})} \right] \tag{17}$$

$$= -I(Z; Z_p) + \mathbb{E}_{p(\mathbf{z}, \mathbf{z}_n)} \left[ \log \sum_{i=1}^{B-1} p(\mathbf{z} \mid \mathbf{z}_n^{(i)}) \right] - \mathbb{E}_{p(\mathbf{z})} \left[ \log p(\mathbf{z}) \right].$$

Using the property of logarithm and Jensen's Inequality, then

$$\mathcal{L}_{\text{InfoNCE}} \geq -I(Z; Z_p) + \mathbb{E}_{p(\mathbf{z}, \mathbf{z}_n)} \left[ \log \sum_{i=1}^{B-1} \frac{p(\mathbf{z} \mid \mathbf{z}_n^{(i)})}{B-1} \right] + \log(B-1) - \mathbb{E}_{p(\mathbf{z})} \left[ \log p(\mathbf{z}) \right]$$

$$\geq -I(Z; Z_p) + \frac{1}{B-1} \sum_{i=1}^{B-1} \mathbb{E}_{p(\mathbf{z}, \mathbf{z}_n)} \left[ \log p(\mathbf{z} \mid \mathbf{z}_n^{(i)}) \right] + \log(B-1) - \mathbb{E}_{p(\mathbf{z})} \left[ \log p(\mathbf{z}) \right]. \tag{18}$$

Note that negative sample $\mathbf{z}_n$s are sampled in same distribution. According to Law of large numbers, we can approximate $\frac{1}{B-1} \sum_{i=1}^{B-1} \mathbb{E}_{p(\mathbf{z}, \mathbf{z}_n)} \left[ \log p(\mathbf{z} \mid \mathbf{z}_n^{(i)}) \right] \approx \mathbb{E}_{p(\mathbf{z}, \mathbf{z}_n)} \left[ \log p(\mathbf{z} \mid \mathbf{z}_n) \right]$.

Therefore, the last inequality Eq. (19) holds.

$$\mathcal{L}_{\text{InfoNCE}} \geq -I(Z; Z_p) + \mathbb{E}_{p(\mathbf{z}, \mathbf{z}_n)} \left[ \log p(\mathbf{z} \mid \mathbf{z}_n) \right] + \log(B-1) - \mathbb{E}_{p(\mathbf{z})} \left[ \log p(\mathbf{z}) \right] \tag{19}$$

$$= -I(Z; Z_p) + I(Z; Z_n) + \log(B-1). \tag{20}$$

$\square$

## A.2 PROOF OF PROPOSITION 2

**Proposition 2.** *For the loss function defined in Eq. (7), the following inequality holds:*

$$\mathcal{L}_{\text{CNO}}^{\gamma} \geq -\frac{1}{\gamma} I(Z; Z_{\text{pos}}) + I(Z; Z_{\text{neg}}) + log(B-1). \tag{21}$$

*Proof.* Compared to Equation Eq. (14), similarity function $g(z, z')$ is replaced with $g_{\gamma}(z, z') = \exp(\frac{f(z,z')}{\gamma\tau}) = \{g(z, z')\}^{\frac{1}{\gamma}}$. Therefore, Equation Eq. (16) can be rewritten as:

$$\mathcal{L}_{\text{CNO}}^{\gamma} \geq -\mathbb{E}_{p(\mathbf{z},\mathbf{z}_p)} \left[ \log\{g(\mathbf{z}, \mathbf{z}_p)\}^{\frac{1}{\gamma}} \right] + \mathbb{E}_{p(\mathbf{z},\mathbf{z}_n)} \left[ \log \sum_{i=1}^{B-1} g(\mathbf{z}, \mathbf{z}_n^{(i)}) \right] \tag{22}$$

$$= -\frac{1}{\gamma}\mathbb{E}_{p(\mathbf{z},\mathbf{z}_p)} \left[ \log g(\mathbf{z}, \mathbf{z}_p) \right] + \mathbb{E}_{p(\mathbf{z},\mathbf{z}_n)} \left[ \log \sum_{i=1}^{B-1} g(\mathbf{z}, \mathbf{z}_n^{(i)}) \right]. \tag{23}$$

Following similar derivations with Appendix A.1, we can simply show that $\mathcal{L}_{\text{CNO}}^{\gamma} \geq -\frac{1}{\gamma}I(Z; Z_p) + I(Z; Z_n) + \log(B-1)$. $\qquad\square$

## B IMPLEMENTATION DETAILS

### B.1 PSEUDOCODE

Detailed algorithm for our sampling method is provided in Algorithm 1.

### B.2 EVALUATION METRICS

- **Image Quality and Prompt Alignment.** To measure the quality and textual relevance of the generated images, we employ a suite of widely-recognized automated metrics.
  - **CLIPScore.** This metric evaluates the semantic consistency between a generated image and its corresponding text prompt by calculating the cosine similarity of their embeddings from a pre-trained CLIP model (Hessel et al., 2021).
  - **PickScore.** We use PickScore (Kirstain et al., 2023), a reward model trained on large-scale human preferences, to assess the overall aesthetic quality and prompt alignment of the images.
  - **Image-Reward.** As a complementary metric, Image-Reward (Xu et al., 2023) is another human-preference-based reward model that provides scores reflecting the general quality of the generated content.
- **Diversity.** To evaluate the intra-prompt diversity of the generated images, we utilize two distinct metrics that capture different aspects of variation.
  - **Vendi Score.** The Vendi Score (Friedman & Dieng, 2023) measures the diversity of a set of samples by analyzing the eigenvalue distribution of their similarity matrix. It provides a holistic assessment of both the variety and balance of the generated images.
  - **Mean Pairwise Similarity (MSS).** This metric directly quantifies the average similarity between all unique pairs of images generated for a single prompt. We first extract image features using the self-supervised descriptor for image copy detection (SSCD) model (Pizzi et al., 2022). Then, we compute the pairwise cosine similarity matrix of these features and calculate the mean of its off-diagonal elements. A lower MSS value indicates higher diversity, as images in the set are, on average, less similar to one another.

### B.3 HYPERPARAMETER SETTINGS

For our main experiments, we use a set of 2K prompts, with each prompt generating a batch of $B$ images. The number of inference steps was set to 50 for Stable Diffusion 1.5 and XL, and 28

Table 2: Model-specific hyperparameters for our proposed method.

| Hyperparameter | Stable Diffusion 1.5 | Stable Diffusion XL | Stable Diffusion 3 |
|---|---|---|---|
| CFG Scale | 6.0 | 6.0 | 7.0 |
| Optimization Steps ($N_{opt}$) | 3 | 3 | 3 |
| Gamma ($\gamma$) | 1.0 | 1.0 | 1.0 |
| Window Size ($w$) | 16 | 16 | 32 |
| Learning Rate ($\eta$) | 0.01 | 0.01 | 0.001 |

for Stable Diffusion 3. The batch size ($B$) was set to 5 for SD1.5 and SD3, and 3 for SDXL. The specific hyperparameters for our proposed method are detailed in Table 2. As shown in the table, most settings are shared across different T2I backbones, highlighting the robustness of our approach to hyperparameter choices.

## C  FURTHER ANALYSES AND DISCUSSIONS

### C.1  STEPWISE MECHANISM OF CONSTRASTIVE NOISE OPTIMIZATION

To clarify the exact role of the attraction coefficient $\gamma$ and the gradient dynamics within our framework, we provide a stepwise breakdown using a minimal batch example.

**Loss Mechanism: Attraction and Repulsion.** Consider a minimal batch of size $B = 2$. Let $\mathcal{L}_1$ denote the loss for the first initial noise vector $\mathbf{z}_T^1$. Given a fixed reference anchor $\hat{\mathbf{z}}_{0|T}^{1,\text{ref}}$ and the similarity function $\text{sim}(\cdot, \cdot)$, the loss decomposes into two distinct forces:

$$\mathcal{L}_1 = -\log \frac{\exp(\text{sim}(\hat{\mathbf{z}}_{0|T}^1, \hat{\mathbf{z}}_{0|T}^{1,\text{ref}})/(\gamma\tau))}{\sum_{j=1}^{2} \exp(\text{sim}(\hat{\mathbf{z}}_{0|T}^1, \hat{\mathbf{z}}_{0|T}^j)/\tau)}$$

$$= \underbrace{-\frac{\text{sim}(\hat{\mathbf{z}}_{0|T}^1, \hat{\mathbf{z}}_{0|T}^{1,\text{ref}})}{\gamma\tau}}_{\text{(A) Attraction}} + \underbrace{\log\left(\exp(\text{sim}(\hat{\mathbf{z}}_{0|T}^1, \hat{\mathbf{z}}_{0|T}^1)/\tau) + \exp(\text{sim}(\hat{\mathbf{z}}_{0|T}^1, \hat{\mathbf{z}}_{0|T}^2)/\tau)\right)}_{\text{(B) Repulsion}}$$

**(A) Attraction.** This term encourages alignment between the current estimate $\hat{\mathbf{z}}_{0|T}^1$ and its fixed reference $\hat{\mathbf{z}}_{0|T}^{1,\text{ref}}$, preserving semantic fidelity to the original concept.

**(B) Repulsion.** It pushes $\hat{\mathbf{z}}_{0|T}^1$ away from other samples in the batch (e.g., $\hat{\mathbf{z}}_{0|T}^2$), enforcing diversity by maximizing semantic distance.

**Role of Gamma ($\gamma$).** The coefficient $\gamma$ serves as a regulator for the fidelity-diversity trade-off by exclusively scaling the attraction term. As derived in Proposition 2, the effective objective is lower-bounded by:

$$\mathcal{L}_{\text{CNO}}^{\gamma} \geq -\frac{1}{\gamma} I(Z; Z_{\text{pos}}) + I(Z; Z_{\text{neg}}) + \log(B - 1).$$

This reveals that $\gamma$ modulates the strength of the positive mutual information. Specifically:

- **Decreasing $\gamma < 1$:** Amplifies the attraction force ($1/\gamma > 1$). This is crucial for larger batch sizes (e.g., $B = 3$), where the cumulative repulsion from $B - 1$ negative pairs can overshadow the single attraction term. A lower $\gamma$ restores balance, preventing the sample from drifting too far from the anchor.
- **Increasing $\gamma > 1$:** Dampens the attraction, allowing the repulsion term to dominate. This promotes greater diversity but risks reducing fidelity.

Practically, we find that setting $\gamma \approx (\tau \ln(B - 1) + 1)^{-1}$ provides a robust baseline for balancing these forces across varying batch sizes.

**Gradient Flow and the Stop-Gradient Strategy.** To optimize $\mathbf{z}_T$ efficiently, we analyze the gradient flow. For the loss $\mathcal{L}_1$ defined above:

- **Active Gradient Flow:** Gradients propagate through $\hat{\mathbf{z}}^1_{0|T}$ in both numerator and denominator, driving it toward the anchor and away from negatives.
- **No Flow to Anchor:** The reference $\hat{\mathbf{z}}^{1,\text{ref}}_{0|T}$ is fixed; thus, no gradients flow through this term, ensuring it remains a stable guidepost.

Crucially, to backpropagate from $\hat{\mathbf{z}}_{0|T}$ to $\mathbf{z}_T$, we utilize a `stopgrad` operation on the diffusion model output $\boldsymbol{\epsilon}_\theta$. By the chain rule:

$$\nabla_{\mathbf{z}_T}\mathcal{L} = \left(\frac{\partial \hat{\mathbf{z}}_{0|T}}{\partial \mathbf{z}_T}\right)^T \nabla_{\hat{\mathbf{z}}_{0|T}}\mathcal{L}$$

$$= \frac{1}{\sqrt{\bar{\alpha}_T}}\left(\mathbf{I} - \sqrt{1-\bar{\alpha}_T}\frac{\partial \boldsymbol{\epsilon}_\theta(\mathbf{z}_T)}{\partial \mathbf{z}_T}\right)^T \nabla_{\hat{\mathbf{z}}_{0|T}}\mathcal{L}.$$

Calculating the full Jacobian $\partial \boldsymbol{\epsilon}_\theta/\partial \mathbf{z}_T$ is computationally prohibitive. By applying `stopgrad`, we set this term to zero, simplifying the update to:

$$\nabla_{\mathbf{z}_T}\mathcal{L} \approx \frac{1}{\sqrt{\bar{\alpha}_T}} \cdot \mathbf{I} \cdot \nabla_{\hat{\mathbf{z}}_{0|T}}\mathcal{L}.$$

This approximation effectively updates the noise directly in the direction of the semantic gradient. This strategy mirrors the effective Jacobian approximation used in Score Distillation Sampling (SDS) (Poole et al., 2023) and NoiseRefine (Ahn et al., 2024), ensuring stable and efficient guidance without the cost of backpropagating through the U-Net.

## C.2 GAMMA EFFECT: STABILIZING OPTIMIZATION PROCESS

To validate the stability of our proposed method, we conduct an ablation study on the hyperparameter $\gamma$ to analyze its impact on output variability. We set $\gamma$ to values of $\{1.0, 0.9, 0.8, 0.7\}$. To ensure that our findings are not contingent on a specific learning rate, we vary the learning rate $\eta$ within the range of $[0.01, 0.02]$. For each setting of $\gamma$, we generate 5 images per prompt, collecting a total of 5K images using SD1.5 model. We then compute evaluation metrics and calculate their sample variance to quantify the statistical variability of the outputs.

The results of this experiment are summarized in Table 3, where we calculate the sample variance of those metrics in $\eta \in [0.01, 0.02]$. We observe a clear **saturation effect**: as $\gamma$ is decreased from 1.0, the variance of the evaluation metrics stabilizes. Specifically, the most significant change in variance occurs when $\gamma$ is reduced from 1.0 to

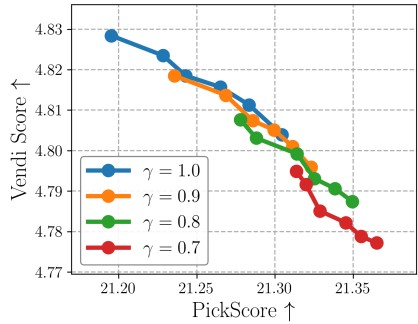

Figure 6: Impact of $\gamma$. The Pareto frontier of PickScore vs. Vendi Score.

0.9. Further decreasing $\gamma$ to 0.8 and 0.7 yields diminishing changes in variance, indicating that the metrics enter a stable regime. For instance, $s^2_{VS}$ exhibits a steady downward trend as $\gamma$ decreases, while the other metrics maintain a relatively consistent level of variance for $\gamma \leq 0.9$.

To further investigate the boundaries of the fidelity-diversity trade-off, we extended our analysis to extreme values of $\gamma$. We evaluated $\gamma = 0.01$ and $\gamma = 100.0$ using the SD1.5 backbone and compared their performance against the standard DDIM sampler and our nominal setting ($\gamma = 1.0$).

The quantitative results are summarized in Table 4. We observe two distinct behaviors at these extremes:

**Strong Attraction** ($\gamma = 0.01$). When $\gamma$ is extremely small, the attraction term becomes overwhelmingly

Table 3: Gamma effect. Subscripts $VS, CS, PS, IR$ mean Vendi Score, CLIPScore, PickScore, Image-Reward, respectively.

| $\gamma$ | Sample variance ($\times 10^{-4}$) | | | |
|---|---|---|---|---|
| | $s^2_{VS}$ | $s^2_{CS}$ | $s^2_{PS}$ | $s^2_{IR}$ |
| 1.0 | 0.076 | 5.49 | 1.54 | 0.60 |
| 0.9 | 0.068 | 2.47 | 1.00 | 0.61 |
| 0.8 | 0.061 | 3.89 | 0.78 | 0.34 |
| 0.7 | 0.050 | 1.61 | 0.41 | 0.11 |

| Method | CLIPScore ↑ | PickScore ↑ | ImReward ↑ | MSS ↓ | Vendi Score ↑ |
|---|---|---|---|---|---|
| DDIM | **31.3940** | 21.3937 | **0.0890** | 0.1485 | 4.7413 |
| Ours ($\gamma = 0.01$) | 31.3934 | **21.3945** | 0.0888 | 0.1485 | 4.7412 |
| Ours ($\gamma = 1.0$) | 31.3424 | 21.2907 | 0.0360 | **0.1276** | **4.7945** |
| Ours ($\gamma = 100.0$) | 31.3764 | 21.2696 | 0.0185 | 0.1279 | 4.7933 |

Table 4: Ablation on extreme gamma values. We compare performance under extreme settings ($\gamma = 0.01$ and $\gamma = 100.0$) against DDIM and our nominal parameter ($\gamma = 1.0$). Extremely low $\gamma$ reverts performance to the DDIM baseline, while extremely high $\gamma$ degrades fidelity without providing the optimal diversity gains achieved at $\gamma = 1.0$.

dominant ($1/\gamma \gg 1$), rigidly anchoring the noise to the initial Tweedie estimate. This effectively suppresses the diversity-inducing repulsion, resulting in metrics nearly identical to the DDIM baseline.

**Weak Attraction ($\gamma = 100.0$).** Conversely, a very large $\gamma$ negates the anchoring force. While this allows for marginal diversity gains, it causes a significant drop in fidelity (PickScore and ImageReward) compared to $\gamma = 1.0$, indicating that the optimization drifts away from semantic alignment without sufficient regularization.

These findings confirm that our nominal value ($\gamma \approx 1.0$) strikes an effective balance, leveraging sufficient attraction to maintain quality while allowing enough repulsive freedom to enhance diversity.

### C.3 Leveraging KL Divergence for Noise Regularization

To further analyze the stability of our method, we investigate the effect of an explicit regularization term. This can be achieved by penalizing the deviation of the optimized noise batch $\{\mathbf{z}_T^i\}_{i=1}^{B}$ from the standard Gaussian prior, $\mathcal{N}(0, \mathbf{I})$, using a Kullback-Leibler (KL) divergence term (Shlens, 2014).

For a single noise tensor $\mathbf{z}_T$, we treat all of its constituent elements as a single population of data points to estimate an underlying distribution. First, we compute the sample mean ($\hat{\mu}$) and sample variance ($\hat{\sigma}^2$) across all $D = C \times H \times W$ elements within the tensor:

$$\hat{\mu} = \frac{1}{D} \sum_{h=1}^{H} \sum_{w=1}^{W} \sum_{c=1}^{C} \mathbf{z}_T[c, h, w], \quad \hat{\sigma}^2 = \frac{1}{D-1} \sum_{h=1}^{H} \sum_{w=1}^{W} \sum_{c=1}^{C} (\mathbf{z}_T[c, h, w] - \hat{\mu})^2.$$

Here, $\mathbf{z}_T[c, h, w]$ represents the pixel value allocated in $c$-th channel and $(h, w)$-position of the latent tensor $\mathbf{z}_T$. These statistics define an estimated univariate Gaussian distribution, $P = \mathcal{N}(\hat{\mu}, \hat{\sigma}^2)$, that characterizes the single noise tensor. We then measure the divergence of this distribution from the standard normal prior, $Q = \mathcal{N}(0, 1)$. The KL divergence for these univariate Gaussian distributions is:

$$D_{KL}(\mathcal{N}(\hat{\mu}, \hat{\sigma}^2) \| \mathcal{N}(0, 1)) = \log \frac{1}{\hat{\sigma}} + \frac{\hat{\sigma}^2 + \hat{\mu}^2}{2} - \frac{1}{2}.$$

By minimizing this KL penalty, we enforce a constraint that encourages the internal statistics of the optimized noise tensor to remain close to those of a standard normal distribution. Integrated algorithm is shown in Algorithm 2.

As shown in Figure 7, incorporating this KL penalty shifts the quality-diversity Pareto frontier to the lower-right, indicating a trade-off towards higher textual fidelity at the cost of lower diversity. Interestingly, we observe an analogous phenomenon in our analysis of the attraction coefficient $\gamma$. As detailed in Section C.2, lowering the value of $\gamma$ similarly shifts the frontier to the lower-right and stabilizes performance; see Figure 6 for details. This parallel suggests that the $\gamma$ coefficient in our contrastive loss implicitly functions as a regularizer, controlling the diversity-fidelity trade-off in a manner similar to an explicit KL divergence penalty.

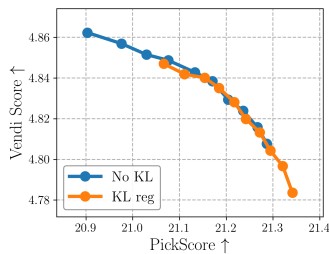

Figure 7: Ablation study on applying Kullback-Leibler divergence. Weight for KL divergence is set as $\lambda = 1000$.

Table 5: Ablation of window size $w$ across prompt types. We compare the effect of full spatial resolution ($w = 64$) versus aggressive downsampling ($w = 1$) on GenEval (simple) and T2I-CompBench (complex).

| Dataset | Setting | CLIPScore ↑ | PickScore ↑ | ImReward ↑ | MSS ↓ | Vendi Score ↑ |
|---|---|---|---|---|---|---|
| GenEval | $w = 64$ | 30.7080 | **21.3857** | **0.0585** | 0.1017 | **4.8376** |
| | $w = 1$ | **30.8394** | 21.3320 | 0.0083 | **0.1016** | 4.8374 |
| T2I-CompBench | $w = 64$ | 30.8639 | **20.0979** | **-0.1697** | 0.1218 | **4.7931** |
| | $w = 1$ | **30.9414** | 20.0856 | -0.2335 | **0.1208** | 4.7879 |

Table 6: Effect of adjusting the initial noise variance. We evaluate the influence of scaling the initial Gaussian prior variance by a factor $\tau$ ($\mathbf{z}_T \sim \mathcal{N}(0, \tau^2\mathbf{I})$). Naively increasing variance fails to improve diversity and degrades quality at higher values, whereas our framework achieves the best diversity-quality trade-off.

| Method | CLIPScore ↑ | PickScore ↑ | ImReward ↑ | MSS ↓ | Vendi Score ↑ |
|---|---|---|---|---|---|
| $\tau = 1.0$ (DDIM) | **31.5041** | 21.4358 | 0.1324 | 0.1620 | 4.7029 |
| $\tau = 1.01$ | 31.4950 | **21.4381** | 0.1597 | 0.1608 | 4.7060 |
| $\tau = 1.025$ | 31.4668 | 21.4313 | 0.1814 | 0.1599 | 4.7090 |
| $\tau = 1.05$ | 31.4469 | 21.3237 | **0.2058** | 0.1621 | 4.7058 |
| $\tau = 1.1$ | 31.4757 | 20.8230 | 0.0817 | 0.1811 | 4.6539 |
| $\tau = 1.15$ | 30.0744 | 19.6513 | -0.6910 | 0.2114 | 4.5561 |
| **Ours** | 31.3424 | 21.2907 | 0.0360 | **0.1276** | **4.7945** |

## C.4 INFLUENCE OF WINDOW SIZE ON PROMPT TYPES

To address the question of whether aggressive downsampling disproportionately affects specific prompt categories, we expanded our ablation study on the window size $w$. While Figure 4 demonstrates the global impact of $w$, here we specifically examine the performance difference between preserving full spatial resolution ($w = 64$) and aggressive downsampling ($w = 1$) across two distinct prompt domains:

- **GenEval** (Ghosh et al., 2023): representing general, simple captions.
- **T2I-CompBench** (Huang et al., 2023): representing complex, compositional prompts that require spatial reasoning.

The comparative results are summarized in Table 5. We observe that for both prompt categories, using $w = 1$ results in degraded performance compared to $w = 64$. This confirms that spatial structure in the initial noise prediction is valuable for optimization. Notably, the degradation in fidelity (PickScore and ImageReward) and diversity (Vendi Score) is observed in both, but the preservation of spatial dimensions ($w = 64$) is particularly critical for maintaining the quality of complex prompts in T2I-CompBench. Aggressive downsampling ($w = 1$) effectively collapses spatial information into a single vector, which hinders the model's ability to optimize for compositional elements that rely on spatial layout. Thus, a moderate to large window size is essential to ensure robustness across varying prompt complexities.

## C.5 IMPACT OF INITIAL NOISE VARIANCE ON DIVERSITY

The initial latent $\mathbf{z}_T$ is standardly sampled from a Gaussian distribution $\mathcal{N}(0, \mathbf{I})$. To investigate whether simply increasing the prior variance promotes diversity, we introduced a scaling factor $\tau$ such that $\mathbf{z}_T \sim \mathcal{N}(0, \tau^2\mathbf{I})$.

As shown in Table 6, slight increases in variance ($\tau \in [1.01, 1.05]$) yield negligible diversity gains, suggesting that small perturbations fail to escape dominant modes. Conversely, larger variances

**Prompt:** *'photo of a cat and a dog running, mountain background."*

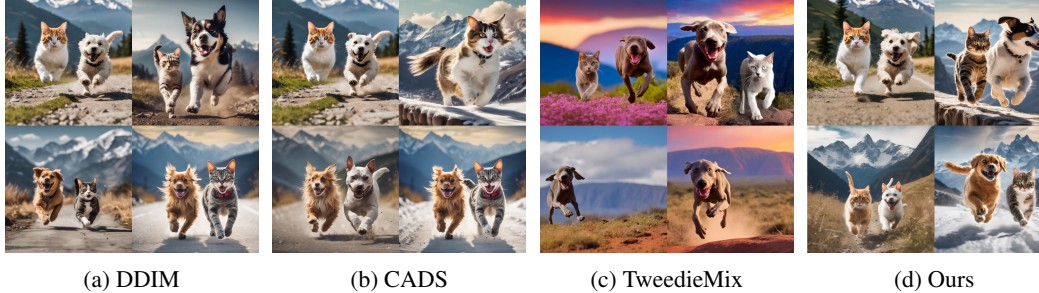

| (a) DDIM | (b) CADS | (c) TweedieMix | (d) Ours |

Figure 8: Qualitative comparison with concept fusion methods. We compare DDIM, CADS, TweedieMix, and ours using the prompt: *"photo of a cat and a dog running, mountain background."* While TweedieMix (c) attempts to fuse concepts, it suffers from low text-image alignment, often failing to generate one of the subjects (*e.g.*, omitting the cat) due to the limitations of personalization-based tuning. In contrast, Ours (d) successfully captures all semantic elements while providing diverse variations.

| Method | CLIPScore ↑ | PickScore ↑ | ImReward ↑ | MSS ↓ | Vendi Score ↑ |
|---|---|---|---|---|---|
| DDIM | **35.9353** | **23.1232** | **1.4163** | 0.2319 | 2.8302 |
| CADS | 35.2270 | 22.7042 | 1.1303 | 0.1860 | 2.8816 |
| TweedieMix | 29.7348 | 20.1795 | -0.4640 | 0.2112 | 2.8531 |
| Ours | 35.3717 | 22.7858 | 1.2632 | **0.1645** | **2.9040** |

Table 7: Quantitative comparison with TweedieMix. We evaluate performance on compositional prompts using personalized concepts. Our method achieves a better balance of alignment (CLIP, PickScore) and diversity (MSS, Vendi) without requiring the pre-training or segmentation steps mandated by TweedieMix.

($\tau \geq 1.1$) cause a sharp decline in fidelity and paradoxically reduce diversity, indicating a failure to converge to meaningful image manifolds. This observation aligns with recent findings that variance inflation deteriorates ODE-based sampling (Um et al., 2025). In contrast, our method achieves superior diversity (Vendi $\approx 4.79$) without such quality collapse, demonstrating that contrastive optimization is far more effective than naive noise scaling.

### C.6 INVESTIGATION OF ALTERNATIVE DIVERSITY APPROACHES

While our primary comparisons focus on zero-shot diversity samplers, the landscape of diversity-enhancing techniques also includes methods based on multi-concept fusion, 3D-aware generation, and counterfactual interventions. In this section, we extend our analysis to these broader families of approaches, incorporating both (i) direct empirical comparison where feasible, and (ii) conceptual discussion for methods whose goals or training regimes differ fundamentally from ours.

To provide a more comprehensive assessment, we therefore evaluate our method against TweedieMix, a recent method that fuses personalized concepts via Tweedie's formula. We conducted experiments using compositional prompts derived from the official personalized concepts provided by TweedieMix (*e.g.*, "dog", "cat", "mountain"). As shown in Table 7 and Figure 8, CNO demonstrates a superior trade-off between diversity and text-image alignment.

We observe that TweedieMix often exhibits degraded fidelity, such as omitting requested subjects (*e.g.*, the cat in Figure 8(c)). This limitation largely stems from the catastrophic forgetting inherent in personalization-based approaches; optimizing for specific concepts can degrade the model's ability to generate general concepts or complex compositions outside the pre-learned set. Furthermore, TweedieMix relies on concept embeddings learned through an additional training stage. This naturally biases generation toward those memorized representations, restricting the model's ability to explore the diverse variations that a zero-shot method like CNO can access. Computationally,

TweedieMix incurs significant overhead as it requires segmenting latents using an external segmentation model (*e.g.*, Text-SAM) prior to fusion. In contrast, our framework requires no extra training or external models, applying directly to arbitrary prompts in a lightweight, one-shot manner.

Beyond multi-concept fusion, several recent approaches advance generative controllability from orthogonal directions, namely 3D consistency, layout reasoning, and counterfactual structure. Diff-Splat (Lin et al., 2025) introduces a differentiable 3D splatting pipeline designed to improve multi-view coherence and geometric accuracy. CoT-lized Diffusion (Liu et al., 2025) incorporates multimodal LLM-based chain-of-thought reasoning into the denoising trajectory to refine spatial arrangements and relational structure. Pan & Bareinboim (2025) propose a causally grounded latent space enabling counterfactual manipulation with invariant factors preserved across interventions. While these approaches significantly enhance compositional fidelity, geometric structure, or causal interpretability, they do not target the diversity–quality trade-off in T2I generation. Their methods typically modify the generative process itself, whereas our approach directly reshapes the initial noise distribution to mitigate mode collapse without altering the sampling trajectory or model architecture, making CNO complementary to these orthogonal research directions.

### C.7 COMPUTATIONAL ANALYSIS

Table 8: Computational cost and performance comparison on Stable Diffusion v1.5.

| Method | Time (sec / batch) $\downarrow$ | PickScore $\uparrow$ | VendiScore $\uparrow$ |
|---|---|---|---|
| DDIM | 11.131 | 21.2398 | 4.8024 |
| Particle Guidance | 11.164 | 21.2164 | 4.8016 |
| CADS | 11.167 | 21.2254 | 4.7964 |
| DiversityPrompt | 18.703 | 21.3067 | 4.7599 |
| Ours (w/o `stopgrad`) | 12.853 | 21.3125 | 4.8010 |
| Ours (with `stopgrad`) | 11.866 | 21.3044 | 4.8039 |

To evaluate the practical efficiency and computational overhead of our proposed method, we conducted a comparative analysis against several baseline and state-of-the-art techniques. All experiments in this section were performed using the **Stable Diffusion v1.5** model.

Our evaluation focuses on the trade-off between computational cost and performance. We generated a total of 5K samples for each method to measure the average time per batch, along with key performance indicators for quality (PickScore) and diversity (VendiScore). The results, summarized in Table 8, provide a clear overview of each method's performance profile.

As presented in Table 8, our approach demonstrates a highly compelling efficiency-performance profile. With an optimization step of $N_{opt} = 3$, our method incurs a modest computational overhead of approximately 5% relative to the standard DDIM sampler.

Despite this, our approach is notably faster and achieves superior metric scores compared to MinorityPrompt. It also remains significantly more efficient than computationally intensive methods like Particle Guidance. Crucially, this slight increase in latency is a highly acceptable trade-off. Our method achieves a unique point on the Pareto frontier of efficiency, quality, and diversity. The combination of high PickScore and VendiScore delivered by our approach represents a state-of-the-art balance unmatched by any other method at any computational cost. This result underscores the practical value of our method, offering a solution that is both powerful and efficient for real-world applications.

**Impact of Batch Size** ($B$). To provide practical guidelines for real-world usage, we further analyzed how the batch size $B$ influences both generation quality and computational overhead. Our empirical findings suggest that the optimal strategy is to set $B$ equal to the number of images intended for generation per prompt (NIPP).

Table 9 illustrates the performance trade-offs. Increasing $B$ from 3 to 5 allows the repulsion term to act on a larger set of samples, pushing them into increasingly diverse directions. Consequently, $B = 5$ achieves a superior fidelity-diversity balance compared to $B = 3$, yielding comparable diversity (Vendi Score) with improved quality metrics (CLIPScore, PickScore, and ImageReward).

Table 9: Impact of batch size $B$ on performance. $B = 5$ yields a better balance, improving quality metrics while maintaining high diversity compared to $B = 3$.

| Method | CLIPScore ↑ | PickScore ↑ | ImReward ↑ | MSS ↓ | Vendi Score ↑ |
|---|---|---|---|---|---|
| DDIM | **31.3940** | **21.3937** | **0.0890** | 0.1485 | 4.7413 |
| Ours ($B = 3$) | 31.3298 | 21.2640 | 0.0238 | **0.1267** | **4.7949** |
| Ours ($B = 5$) | 31.3424 | 21.2907 | 0.0360 | **0.1276** | 4.7945 |

Table 10: Influence of batch size $B$ on computations. We report Peak Memory (MiB) and Time (sec/batch). Our method remains efficient even at larger batch sizes compared to DiversityPrompt.

| Method | $B = 3$ | | $B = 5$ | |
|---|---|---|---|---|
| | Memory (MiB) ↓ | Time (sec/batch) ↓ | Memory (MiB) ↓ | Time (sec/batch) ↓ |
| DDIM | 7694 | **8.091** | 10978 | **12.778** |
| Ours | **7672** | 8.809 | **10828** | 13.427 |
| DiversityPrompt | 13070 | 15.063 | 19006 | 21.631 |

We generally do not observe stagnation in diversity improvement up to the NIPP limit; thus, utilizing the largest feasible $B$ is recommended.

Regarding computational cost, while increasing $B$ naturally raises memory usage and inference time, the overhead for our method remains relatively marginal. Crucially, even at $B = 5$, our approach is significantly more efficient than iterative baselines such as DiversityPrompt, which incurs nearly double the memory and time cost (see Table 10). This efficiency stems from our one-shot optimization of the initial noise, avoiding the heavy cost of iterative interventions during the sampling process.

## C.8 LIMITATIONS AND FUTURE WORK

Despite its effectiveness in mitigating mode collapse, our method – like other diversity-focused approaches – exhibits limitations when handling prompts that require strong compositional grounding. In scenarios involving specific spatial relations or interactions, the optimization for diversity can occasionally compromise text-image alignment. For instance, as illustrated in Figure 9, while standard sampling successfully captures specific relational details (*e.g.*, a dog *holding* a frisbee), diversity methods including CADS, PG, and our method (at $\gamma = 1.0$) may struggle to fully preserve this structural coherence.

**Prompt:** *"A dog holding a yellow frisbee in its mouth."*

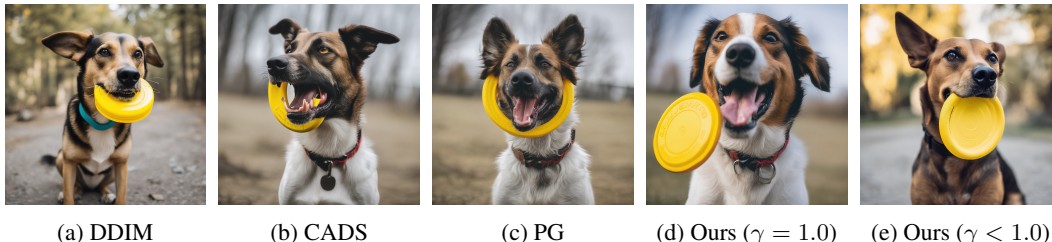

| (a) DDIM | (b) CADS | (c) PG | (d) Ours ($\gamma = 1.0$) | (e) Ours ($\gamma < 1.0$) |
|---|---|---|---|---|

Figure 9: Failure case analysis. We illustrate a scenario requiring strong compositional grounding (specifically, the relation of *holding*). While the standard DDIM sampler (a) correctly captures the relational structure, diversity-enhancing methods (b-d) – including Ours with the nominal $\gamma = 1.0$ – may struggle to maintain this fine-grained alignment. However, unlike other baselines, our framework provides a remedy: by lowering the attraction coefficient ($\gamma < 1.0$), we can recover the correct semantic alignment (e) by trading off a small degree of diversity.

Crucially, however, our framework offers a novel advantage to address this challenge: the attraction coefficient $\gamma$. Unlike existing baselines where the fidelity-diversity trade-off is often rigid, our method enables dynamic control via $\gamma$. As demonstrated in Figure 9(e), by lowering $\gamma$ below 1.0, we can intensify the anchoring force toward the initial Tweedie estimate. This effectively recovers the correct compositional alignment (e.g., the holding relation) at the cost of a modest reduction in diversity. This capability confirms that $\gamma$ functions as an effective fidelity controller, allowing users to flexibly navigate failure cases that are otherwise difficult to resolve in competing frameworks.

For future work, while our method focuses on the initial noise $\mathbf{z}_T$, we believe that applying a similar optimization strategy to intermediate latents $\mathbf{z}_t$ (where $t < T$) could be a promising avenue for further enhancing diversity for a single prompt. Some studies have explored optimizing these intermediate latents to generate images with high fidelity to complex textual conditions (Wallace et al. 2023 ; Ding et al. 2024). The effectiveness of such an approach may depend on the model and the degree to which its Tweedie prediction is already structured to reflect the semantic content of the input prompt at intermediate timesteps. This direction may warrant deeper investigation on our approach.

# D  ADDITIONAL EXPERIMENTAL RESULTS

## D.1  PERFORMANCE ON ACCELERATED MODELS

| Model | Method | CLIPScore ↑ | PickScore ↑ | ImageReward ↑ | MSS ↓ | Vendi Score ↑ |
|---|---|---|---|---|---|---|
| SDXL-Lightning (4-step) | DDIM | **31.5536** | **22.6598** | **0.7231** | 0.2865 | 2.7470 |
| | CADS | 31.4425 | 22.5767 | 0.6876 | 0.2674 | 2.7749 |
| | Ours | 31.4474 | 22.5659 | 0.6740 | **0.2289** | **2.8258** |
| FLUX-1-Schnell (4-step) | FM-ODE | **32.1012** | **22.7411** | **1.0499** | 0.3012 | 2.7220 |
| | CADS | 31.7153 | 22.5431 | 0.8622 | 0.2287 | 2.8250 |
| | Ours | 32.0664 | 22.6137 | 1.0070 | **0.2231** | **2.8316** |

Table 11: Quantitative comparison on few-step accelerated models. We evaluate performance on **SDXL-Lightning** and **FLUX-1-Schnell** under a 4-step inference setting. Our method is compared against DDIM, CADS, and Flow-Matching ODE (FM-ODE), the default sampler used in FLUX. The results indicate that our approach maintains robust diversity improvements (higher Vendi Score, lower MSS) even in aggressive few-step regimes where iterative interventions like CADS are less effective.

**Prompt:** *"An astronaut on the moon"*

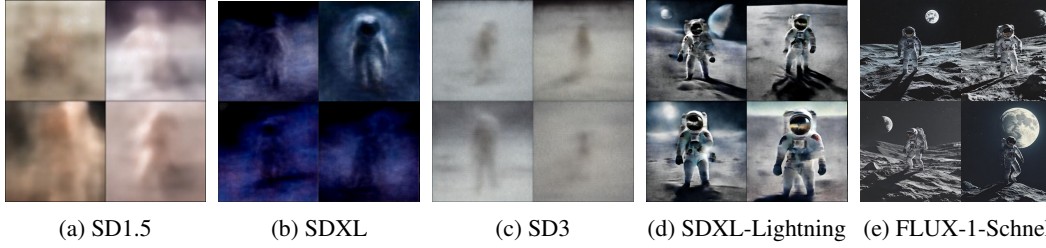

(a) SD1.5    (b) SDXL    (c) SD3   (d) SDXL-Lightning (e) FLUX-1-Schnell

Figure 10: Visualization of initial Tweedie estimates ($\hat{\mathbf{z}}_{0|T}$) across different backbones. We compare the one-step denoised predictions from the initial Gaussian noise $\mathbf{z}_T$.

Few-step accelerated models, while computationally efficient, are prone to severe mode collapse due to limited stochasticity in their shortened trajectories. To validate the robustness of our approach in this regime, we evaluated CNO on two representative accelerated models: **SDXL-Lightning** (4-step distilled diffusion) and **FLUX-1-Schnell** (4-step rectified flow).

As summarized in Table 11, CNO delivers consistent improvements in diversity metrics (Vendi Score and MSS) across both architectures. Notably, we observe that iterative diversity samplers like CADS yield limited gains in this 4-step setting. This performance gap stems from a fundamental operational difference: iterative methods rely on accumulating guidance over many timesteps, a mechanism that becomes ineffective when the sampling trajectory is drastically condensed.

In contrast, CNO performs a one-shot optimization of the initial noise $\mathbf{z}_T$ prior to sampling, making it independent of the inference step count. Crucially, we found that accelerated models tend to produce significantly clearer and more deterministic Tweedie predictions ($\hat{\mathbf{z}}_{0|T}$) at the initial timestep compared to standard many-step models (see Figure 10). This characteristic renders the initial noise optimization particularly effective, as the contrastive gradients derived from these sharp Tweedie estimates are highly semantically meaningful. These results confirm that our framework is model-agnostic and robust even under aggressive sampling acceleration.

## D.2 USER STUDY RESULTS

|  | Alignment & Quality (%) | | Diversity & Creativity (%) | |
| --- | --- | --- | --- | --- |
|  | Baseline | CNO (Ours) | Baseline | CNO (Ours) |
| vs. CADS | 21.29 | **78.71** | 16.45 | **83.55** |
| vs. PG | 41.29 | **58.71** | 21.94 | **78.06** |

Table 12: Human preference evaluation. We compare CNO against CADS and Particle Guidance (PG). The values represent the percentage of user preference. CNO demonstrates a strong advantage in both image quality and generation diversity.

To complement our quantitative evaluation, we conducted a human preference study comparing CNO against two diversity-focused baselines: CADS (Sadat et al., 2024) and Particle Guidance (PG) (Corso et al., 2024). A total of 32 participants evaluated 30 randomly ordered image pairs, each generated using identical prompts and seeds. For each pair, participants selected the preferred sample according to two criteria: (i) *Alignment & Quality* (text adherence and visual fidelity) and (ii) *Diversity & Creativity* (semantic distinctiveness and variation).

As shown in Table 12, CNO is consistently preferred over both baselines. Notably, against CADS, our method achieved preference rates of **78.71%** for quality and **83.55%** for diversity. A similar trend is observed against PG, confirming that the quantitative gains of CNO translate into perceptibly superior generation quality and diversity.

## D.3 EXPERIMENTS ON DISTINCT PROMPT DOMAINS

Table 13: Quantitative evaluation on the GenEval benchmark.

| Method | CLIPScore ↑ | PickScore ↑ | ImReward ↑ | MSS ↓ | Vendi Score ↑ |
| --- | --- | --- | --- | --- | --- |
| DDIM | **32.0443** | **21.7006** | **-0.1422** | 0.1389 | 4.7550 |
| CADS | 31.6739 | 21.4260 | -0.3117 | 0.1020 | 4.8433 |
| Ours | 31.6738 | 21.4923 | -0.2841 | **0.1006** | **4.8508** |

To verify that our method's effectiveness is not limited to the daily scenes typical of MS-COCO, we extended our evaluation to the **GenEval** benchmark (Ghosh et al., 2023). GenEval is designed to test compositional reasoning and includes a diverse set of prompts distinct from standard captioning datasets. We evaluated Stable Diffusion v1.5 using this benchmark, comparing CNO against DDIM and CADS.

The quantitative results are presented in Table 13. Consistent with our main findings, CNO achieves the highest diversity scores (Vendi Score and MSS), outperforming both the standard sampler and the baseline diversity method (CADS). Crucially, while maintaining superior diversity, CNO also

retains higher image quality compared to CADS, as evidenced by higher PickScore and ImageReward values. This demonstrates that the robustness of our contrastive noise optimization extends beyond specific datasets and effectively generalizes to diverse textual domains.

### D.4 ADDITIONAL GENERATED SAMPLES

Figure 11 and Figure 12 represent that CNO shows high image quality and textual fidelity compared to DDIM, CADS, and Particle Guidance which the hyperparameters of our method are equivalent to Table 2.

## E USE OF LARGE LANGUAGE MODELS

We used Large Language Models (LLMs) to aid in the verbal refinement and polishing of our paper. This usage was only limited to improving readability and fixing some grammar errors. The core research, including the formulation of our method, experimental design, and analysis of results, was conducted solely by the authors.

---

**Algorithm 1** Diverse T2I Generation via Contrastive Noise Optimization

---

**Inputs:** Text embedding $c$, batch size $B$, optimization steps $N_{\text{opt}}$, learning rate $\eta$, temperature $\tau$, attraction coefficient $\gamma$, window size $w$
**Outputs:** $\{\mathbf{x}_0^i\}_{i=1}^B$, a batch of diverse images.

---

1: **Models:** Diffusion model $\epsilon_\theta$, DDIM sampler $\mathcal{D}_{\text{DDIM}}$
2: Initialize a batch of trainable noise vectors $\mathbf{Z}_T = \{\mathbf{z}_T^i\}_{i=1}^B \sim \mathcal{N}(0, \mathbf{I})$
3: Let $\mathbf{Z}_T^{\text{ref}} \leftarrow \mathbf{Z}_T$
4: **for** $n = 1$ **to** $N_{\text{opt}}$ **do**
5:     **for** $i = 1$ **to** $B$ **do**
6:         $\hat{\mathbf{z}}_{0|T}^i \leftarrow \frac{1}{\sqrt{\bar{\alpha}_T}}(\mathbf{z}_T^i - \sqrt{1 - \bar{\alpha}_T}\,\texttt{stopgrad}\{\epsilon_\theta(\mathbf{z}_T^i, T, c)\})$
7:         **if** $n = 1$ **then**
8:             $\hat{\mathbf{z}}_{0|T}^{i,\text{ref}} \leftarrow \hat{\mathbf{z}}_{0|T}^i$
9:         **end if**
10:        $\hat{\mathbf{z}}_{0|T}^i, \hat{\mathbf{z}}_{0|T}^{i,\text{ref}} \leftarrow \text{DownSample}(\hat{\mathbf{z}}_{0|T}^i; w), \;\; \text{DownSample}(\hat{\mathbf{z}}_{0|T}^{i,\text{ref}}; w)$
11:        $\hat{\mathbf{z}}_{0|T}^i, \hat{\mathbf{z}}_{0|T}^{i,\text{ref}} \leftarrow \text{Normalize}(\hat{\mathbf{z}}_{0|T}^i), \;\; \text{Normalize}(\hat{\mathbf{z}}_{0|T}^{i,\text{ref}})$
12:     **end for**
13:     $\mathcal{L}_{\text{CNO}}^\gamma := \frac{1}{B}\sum_{i=1}^B \left[ -\log\left( \frac{\exp(f(\hat{\mathbf{z}}_{0|T}^i, \hat{\mathbf{z}}_{0|T}^{i,\text{ref}})/(\gamma\tau))}{\sum_{j=1}^B \exp(f(\hat{\mathbf{z}}_{0|T}^i, \hat{\mathbf{z}}_{0|T}^j)/\tau)} \right) \right]$
14:     $\mathbf{Z}_T \leftarrow \mathbf{Z}_T - \eta \cdot \nabla_{\mathbf{Z}_T}\mathcal{L}_{\text{CNO}}^\gamma$
15: **end for**
16: $\{\mathbf{z}_0^i\}_{i=1}^B \leftarrow \mathcal{D}_{\text{DDIM}}(\mathbf{Z}_T, c)$
17: $\{\mathbf{x}_0^i\}_{i=1}^B \leftarrow \text{Decode}(\{\mathbf{z}_0^i\}_{i=1}^B)$
18: **return** $\{\mathbf{x}_0^i\}_{i=1}^B$

---

**Algorithm 2** Contrastive Noise Optimization with KL Regularization

---

**Inputs:** Text embedding $c$, batch size $B$, optimization steps $N_{\text{opt}}$, learning rate $\eta$, temperature $\tau$, attraction coefficient $\gamma$, window size $w$, KL divergence weight $\lambda$
**Outputs:** $\{\mathbf{x}_0^i\}_{i=1}^B$, a batch of diverse images.

---

1: **Models:** Diffusion model $\epsilon_\theta$, DDIM sampler $\mathcal{D}_{\text{DDIM}}$
2: Let $D$ be the number of pixels in a single noise tensor (channel $C \times$ height $H \times$ width $W$)
3: Initialize a batch of trainable noise vectors $\mathbf{Z}_T = \{\mathbf{z}_T^i\}_{i=1}^B \sim \mathcal{N}(0, \mathbf{I})$
4: Let $\mathbf{Z}_T^{\text{ref}} \leftarrow \mathbf{Z}_T$
5: **for** $n = 1$ **to** $N_{\text{opt}}$ **do**
6:     **for** $i = 1$ **to** $B$ **do**
7:         $\hat{\mathbf{z}}_{0|T}^i \leftarrow \frac{1}{\sqrt{\bar{\alpha}_T}}(\mathbf{z}_T^i - \sqrt{1 - \bar{\alpha}_T}\,\texttt{stopgrad}\{\epsilon_\theta(\mathbf{z}_T^i, T, c)\})$
8:         **if** $n = 1$ **then**
9:             $\hat{\mathbf{z}}_{0|T}^{i,\text{ref}} \leftarrow \hat{\mathbf{z}}_{0|T}^i$
10:        **end if**
11:        $\hat{\mathbf{z}}_{0|T}^i, \hat{\mathbf{z}}_{0|T}^{i,\text{ref}} \leftarrow \text{DownSample}(\hat{\mathbf{z}}_{0|T}^i; w), \;\; \text{DownSample}(\hat{\mathbf{z}}_{0|T}^{i,\text{ref}}; w)$
12:        $\hat{\mathbf{z}}_{0|T}^i, \hat{\mathbf{z}}_{0|T}^{i,\text{ref}} \leftarrow \text{Normalize}(\hat{\mathbf{z}}_{0|T}^i), \;\; \text{Normalize}(\hat{\mathbf{z}}_{0|T}^{i,\text{ref}})$
13:     **end for**
14:     $\mathcal{L}_{\text{CNO}}^\gamma := \frac{1}{B}\sum_{i=1}^B \left[ -\log\left( \frac{\exp(f(\hat{\mathbf{z}}_{0|T}^i, \hat{\mathbf{z}}_{0|T}^{i,\text{ref}})/(\gamma\tau))}{\sum_{j=1}^B \exp(f(\hat{\mathbf{z}}_{0|T}^i, \hat{\mathbf{z}}_{0|T}^j)/\tau)} \right) \right]$
15:     $\hat{\mu}_i \leftarrow \frac{1}{D}\sum_{h=1}^H \sum_{w=1}^W \sum_{c=1}^C \mathbf{z}_T[c, h, w] \quad \text{for } i = 1, \ldots, B$
16:     $\hat{\sigma}_i^2 \leftarrow \frac{1}{D-1}\sum_{h=1}^H \sum_{w=1}^W \sum_{c=1}^C (\mathbf{z}_T[c, h, w] - \hat{\mu})^2 \quad \text{for } i = 1, \ldots, B$
17:     $\mathcal{L}_{\text{KL}} := \frac{1}{B}\sum_{i=1}^B \left[ \log\frac{1}{\hat{\sigma}_i} + \frac{\hat{\sigma}_i^2 + \hat{\mu}_i^2}{2} - \frac{1}{2} \right]$
18:     $\mathcal{L}_{\text{total}} \leftarrow \mathcal{L}_{\text{CNO}}^\gamma + $ $\lambda\mathcal{L}_{\text{KL}}$
19:     $\mathbf{Z}_T \leftarrow \mathbf{Z}_T - \eta \cdot \nabla_{\mathbf{Z}_T}\mathcal{L}_{\text{total}}$
20: **end for**
21: $\{\mathbf{z}_0^i\}_{i=1}^B \leftarrow \mathcal{D}_{\text{DDIM}}(\mathbf{Z}_T, c)$
22: $\{\mathbf{x}_0^i\}_{i=1}^B \leftarrow \text{Decode}(\{\mathbf{z}_0^i\}_{i=1}^B)$
23: **return** $\{\mathbf{x}_0^i\}_{i=1}^B$

---

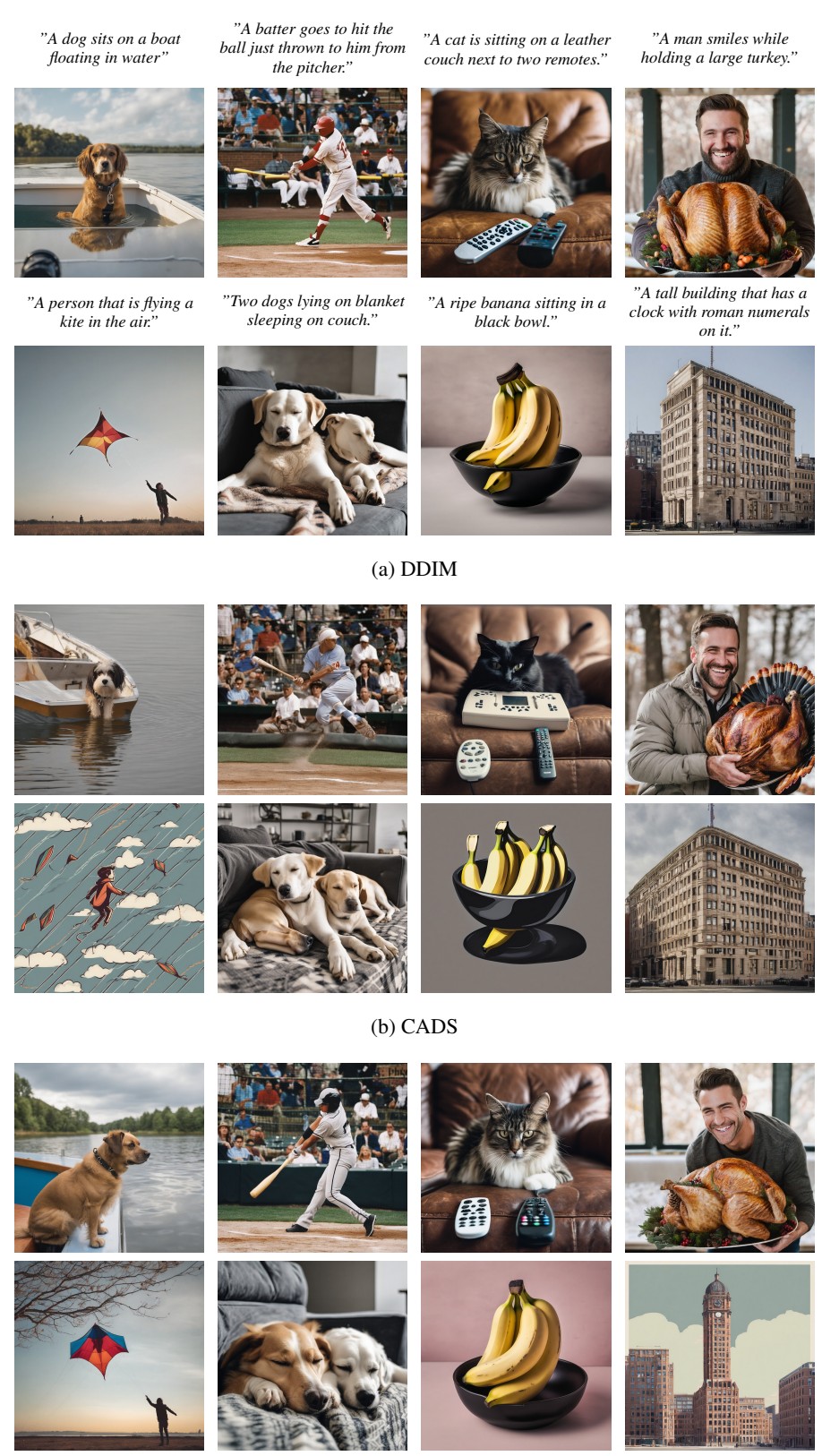

Figure 11: Qualitative comparison of images generated from the same set of text prompts by (a) DDIM, (b) CADS and (c) our proposed method. Images with the same position in individual grids share the same prompt and seed.

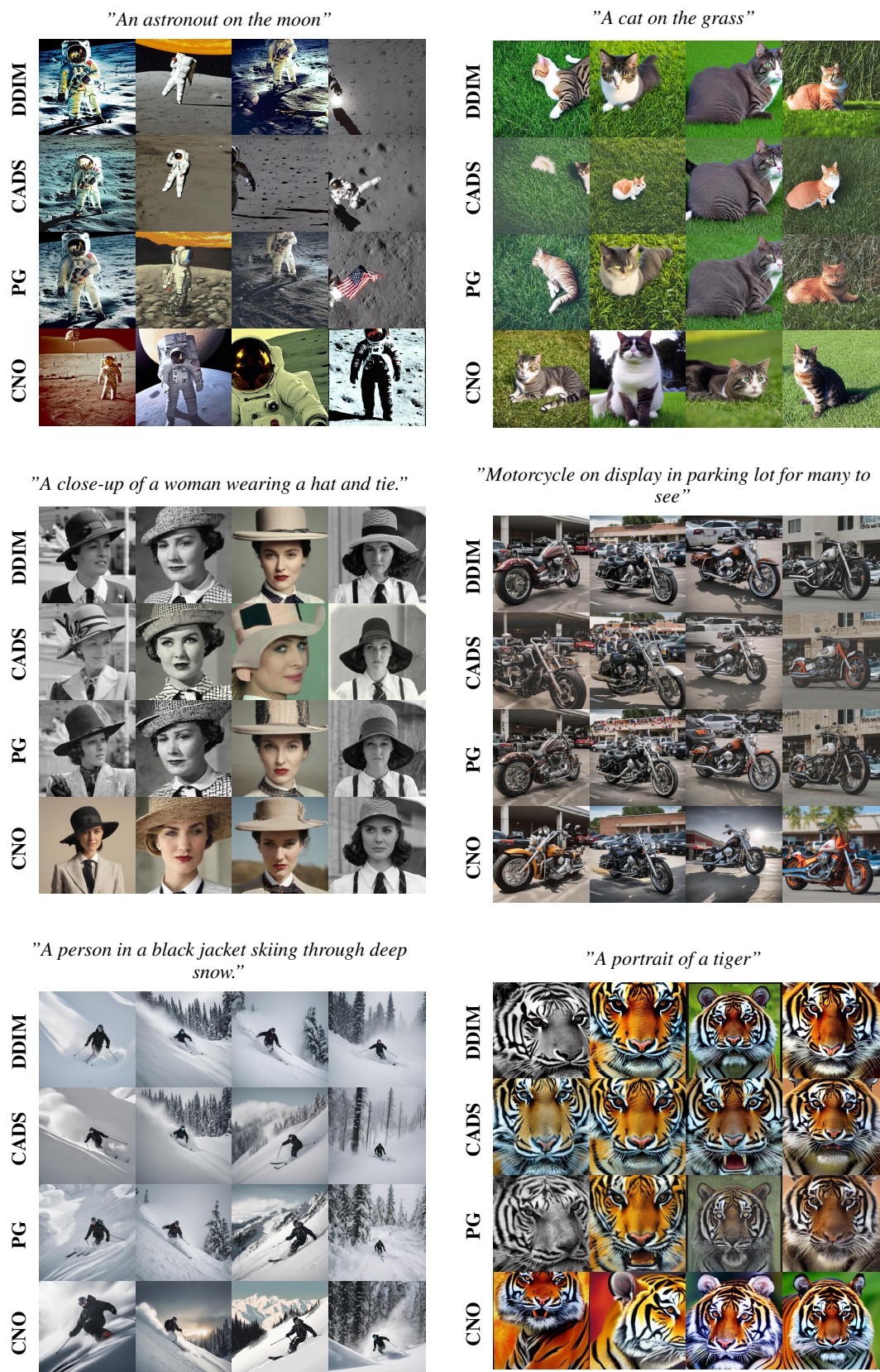

Figure 12: Additional results of synthesized images across different diffusion sampling methods. Each row represents a specific diversity method: DDIM, CADS, Particle Guidance (PG), and our proposed CNO. CNO consistently demonstrates high diversity compared to baseline methods.

