# OpenReview forum: "Diverse Text-to-Image Generation via Contrastive Noise Optimization"
_ICLR.cc/2026/Conference — ICLR 2026 Poster_

### Official Review · Reviewer_ZwsY · 2025-10-19

**Soundness:** 3
**Presentation:** 3
**Contribution:** 3
**Rating:** 6
**Confidence:** 3

**Summary:**

The paper addresses T2I diffusion models’ mode collapse (limited diversity under strong text guidance) via Contrastive Noise Optimization (CNO). It optimizes initial noise latents (before DDIM sampling) using a contrastive loss in Tweedie denoised space: repelling batch samples to boost diversity, while anchoring each to its unoptimized version to preserve fidelity. A γ coefficient balances repulsion/attraction for stability. It achieves a superior quality-diversity Pareto frontier, enabling creative applications (e.g., diverse concept generation) without complex tuning, advancing practical T2I usability.

**Strengths:**

1. It innovatively addresses T2I diversity by optimizing initial noise (not intermediate latents) via Contrastive Noise Optimization (CNO)—a contrastive loss in Tweedie denoised space (not latent space) with repulsion (batch sample separation) and attraction (anchoring to unoptimized noise). This differs from CADS (text embedding perturbation) and PG (intermediate latent repulsion), solving diversity at the source while avoiding hyperparameter fragility.

2. It sets a superior quality-diversity Pareto frontier, with 5% overhead vs. DDIM but outperforming costly baselines (DiversityPrompt). Enabling diverse, high-fidelity generation (e.g., creative concept design) without complex tuning, it advances practical T2I usability and guides future noise-optimization research.

3. Rigorous validation on SD1.5/SDXL/SD3 shows top Vendi Score (diversity) and PickScore (quality). From Fig.3, the CNO is outperforming DDIM/CADS/PG consistently. Such observations are also true after reading the supplementary.

**Weaknesses:**

1. Experiments only use MS-COCO validation prompts (daily scenes) but lack other-domain prompts. Maybe the authors could consider to include other kind of prompt datasets to broaden this setup.

2. The CNO is only tested on the SD series models, I wonder would it be applicable to other structure based T2I models? Flux, DeepFloyd, etc.

**Questions:**

refer to the above.

---

> ### Author Response · Authors · 2025-11-20
> **Official Comment by Authors**
>
> We appreciate Reviewer `ZwsY` for the valuable feedback and constructive suggestions. Below, we provide point-by-point responses to address your concerns.
>
> ---
>
> > **W1.** Experiments only use MS-COCO validation prompts (daily scenes) but lack other-domain prompts. Maybe the authors could consider to include other kind of prompt datasets to broaden this setup.
>
> To address your concern, we conducted additional experiments using a prompt set from a domain different from MS-COCO. Specifically, we evaluated SD1.5 on the GenEval prompt set [1] and compared our method against the baselines. The results are summarized in the table below.
>
> |  | CLIPScore ↑ | PickScore ↑ | ImReward ↑ | MSS ↓ | Vendi Score ↑ |
> |------|:-----------:|:------------:|:-----------:|:-----:|:--------:|
> | **DDIM** | **32.0443** | **21.7006** | **-0.1422** | 0.1389 | 4.7550 |
> | **CADS** | 31.6739 | 21.4260 | -0.3117 | 0.1020 | 4.8433 |
> | **CNO**  | 31.6738 | 21.4923 | -0.2841 | **0.1006** | **4.8508** |
>
> Observe that our framework consistently outperforms the baselines even in this distinct prompt domain, demonstrating that our approach is robust to the choice of text prompts. We have included the above results in our revision; see Section D.3 for details (highlighted in blue).
>
> ---
>
> > **W2.** The CNO is only tested on the SD series models, I wonder would it be applicable to other structure based T2I models? Flux, DeepFloyd, etc.
>
> In response to this comment, we also examined the applicability of our method beyond the SD family. Specifically, we evaluated our approach using FLUX-1-Schnell (4-step), a representative T2I model distinct from SD-based architectures, and compared our performance with baselines. The results are summarized in the following table:
>
> |  | CLIPScore ↑ | PickScore ↑ | ImReward ↑ | MSS ↓ | Vendi Score ↑ |
> |:-------|:-----------:|:------------:|:-----------:|:------:|:---------------:|
> | **FM-ODE** | **32.1012** | **22.7411** | **1.0499** | 0.3012 | 2.7220 |
> | **CADS** | 31.7153 | 22.5431 | 0.8622 | 0.2287 | 2.8250 |
> | **CNO** | 32.0664 | 22.6137 | 1.0070 | **0.2231** | **2.8316** |
>
> Again, our method demonstrates consistent improvements over the baselines, indicating that CNO is not tied to specific diffusion architectures and is broadly applicable across different T2I frameworks. Our revised manuscript now includes these additional FLUX experiments; see Section D.1 for details (highlighted in blue).
>
> ---
>
> **References**
>
> [1] GenEval: An Object-Focused Framework for Evaluating Text-to-Image Alignment, NeurIPS 2023.

---

### Official Review · Reviewer_A3GX · 2025-10-24

**Soundness:** 3
**Presentation:** 3
**Contribution:** 3
**Rating:** 6
**Confidence:** 4

**Summary:**

This paper addresses the pervasive problem of limited diversity in text-to-image (T2I) diffusion models, particularly under strong text guidance. The authors propose Contrastive Noise Optimization (CNO), a pre-processing framework that directly optimizes the batch of initial noise latents to encourage diversity, applying a contrastive InfoNCE loss in the denoised Tweedie space. The approach balances a repulsion term (encouraging diversity) with an attraction to the original sample (preserving semantic fidelity), and introduces a coefficient $\gamma$ to stabilize this trade-off. Theoretical analysis and comprehensive experiments across multiple diffusion backbones demonstrate improvements in both image diversity and quality.

**Strengths:**

1. The paper addresses the diversity challenge at its root by optimizing initial noise, rather than intervening mid-sampling or in text embedding space. This leads to a simple, efficient method that is model-agnostic and does not require modifications to the diffusion backbone.
2. The authors extend the mutual information perspective of InfoNCE to simultaneously account for positive and negative pairs (Section 4.3), providing formal justification for how diversity and fidelity are balanced. They further detail how the $\gamma$ parameter scales this effect.
3. Substantial empirical evidence is provided, including Table 1 and Figure 3, benchmarking against DDIM, Particle Guidance (PG), CADS, and DiversityPrompt across three major diffusion models (SD 1.5, XL, 3). The proposed method consistently outperforms baselines in key diversity metrics (Vendi Score, MSS) and maintains or improves quality/fidelity (PickScore, CLIPScore).
4. The exposition is clear, diagrams such as Figure 2 provide an intuitive grasp of the architecture/mechanism, and the methodology is described in practical, replicable pseudocode (Algorithm 1, Appendix B.1).
5. The method shows low sensitivity to hyperparameter choices (Section 5, Table 2), making it appealing for real-world adoption.

**Weaknesses:**

1. While the theoretical analysis is a strength, there is a (potentially confusing) notational inconsistency when switching between $\mathcal{L}{\text{CNO}}$, $\mathcal{L}{\text{InfoNCE}}$, and $\mathcal{L}_{\text{CNO}}^\gamma$ across pages 5-6. The exact role of $\gamma$ in the contrastive loss numerator/denominator, and how gradients are stopped or propagated through each component, could be clarified further. For readers less familiar with contrastive objectives applied to image generation, a stepwise, worked example would reduce ambiguity.
2. Section 5.1 references an ablation in Figure 4 testing $w\in{4, 8, 16, 64}$, yet the impact on specific image types, prompt complexity, or semantic regions is not deeply probed. There is no discussion of whether aggressive downsampling disproportionately harms long-tail or compositional prompts.
3. While automated metrics (Vendi, CLIPScore, etc.) are robust and reflect recent trends, there is a notable absence of human evaluation for subjective diversity, creativity, or aesthetic ranking beyond PickScore. As text-to-image generation is ultimately a user-facing task, such a study would strongly solidify the real-world value of the method.

**Questions:**

1. Could the authors provide additional quantitative and qualitative results on how extreme values of $\gamma$ (especially $<<1$ and $>>1$) impact the fidelity/diversity trade-off? Are there prompts or settings where the method consistently fails or collapses into poor outputs?
2. What are the empirical impacts and best practices for choosing the batch size $B$ in real-world usage? Is there a threshold beyond which diversity improvements stagnate or degrade?
3. Is the optimization of the initial noise vectors consistently stable across different diffusion model architectures, or are there architectures/settings for which CNO performs suboptimally?

---

> ### Author Response · Authors · 2025-11-20
> **Official Comment by Authors (1)**
>
> We thank Reviewer `A3GX` for the thoughtful and constructive feedback. Below we provide point-by-point responses to address the questions and concerns raised in the review.
>
> ---
>
> > **W1-1.** While the theoretical analysis is a strength, there is a (potentially confusing) notational inconsistency when switching between $\mathcal{L}\_{\text{CNO}}$, $\mathcal{L}\_{\text{InfoNCE}}$, and $\mathcal{L}\_{\text{CNO}}^{\gamma}$ across pages 5-6.
>
> Thank you for pointing this out. We have revised the manuscript to enforce consistent notation across all variants of the objective, including $\mathcal{L}\_{\text{InfoNCE}}$, $\mathcal{L}\_{\text{CNO}}$, and $\mathcal{L}\_{\text{CNO}}^{\gamma}$. All expressions referring to these losses are now unified and use a single, coherent notation throughout the paper (see marked revisions).
>
> ---
>
>
> > **W1-2.** The exact role of gamma in the contrastive loss numerator/denominator, and how gradients are stopped or propagated through each component, could be clarified further. For readers less familiar with contrastive objectives applied to image generation, a stepwise, worked example would reduce ambiguity.
>
>
> Thank you for the constructive feedback. We agree that a more explicit, step-by-step explanation of the loss and its gradient flow improves clarity. Accordingly, we have added the following clarification in Section C.1.
>
> If you encounter raw LaTeX code (e.g., \mathbf{...}) instead of properly rendered equations, please kindly refresh the page. This is a known intermittent issue on OpenReview.
>
> **1. Loss Mechanism: Attraction and Repulsion**
>
> CNO refines initial latents through the interplay of two complementary forces: attraction, which preserves fidelity to the reference, and repulsion, which promotes diversity. To illustrate this, consider the minimal example with batch size $B=2$. Let $\mathcal{L}\_1$ denote the loss for the initial latent $\mathbf{z}\_T^{1}$, where the overall objective is $\mathcal{L}_{\text{CNO}}^{\gamma} = \mathcal{L}\_1 + \mathcal{L}\_2$ (Eq. (7)). Given the fixed anchor $\hat{\mathbf{z}}\_{0|T}^{1,\text{ref}}$, and letting $\text{sim}(\cdot, \cdot)$ denote the similarity function (e.g., cosine similarity), the loss $\mathcal{L}\_1$ decomposes into attraction and repulsion terms as:
> \begin{align*}
> \mathcal{L}\_1 &= - \log \frac{  \exp(\text{sim}(\hat{\mathbf{z}}\_{0|T}^{1}, \hat{\mathbf{z}}\_{0|T}^{1,\text{ref}})/ (\gamma\tau) )  }{   \exp(\text{sim}(\hat{\mathbf{z}}\_{0|T}^{1}, \hat{\mathbf{z}}\_{0|T}^{1})/\tau)  + \exp(\text{sim}(\hat{\mathbf{z}}\_{0|T}^{1}, \hat{\mathbf{z}}\_{0|T}^{2})/\tau) } \newline
> & = \underbrace{- \frac{\text{sim}(\hat{\mathbf{z}}\_{0|T}^{1}, \hat{\mathbf{z}}\_{0|T}^{1,\text{ref}})}{\gamma\tau}}\_{\text{(A) Attraction}} + \underbrace{\log \left( \exp(\text{sim}(\hat{\mathbf{z}}\_{0|T}^{1}, \hat{\mathbf{z}}\_{0|T}^{1})/\tau)  + \exp(\text{sim}(\hat{\mathbf{z}}\_{0|T}^{1}, \hat{\mathbf{z}}\_{0|T}^{2})/\tau) \right)}\_{\text{(B) Repulsion}}
> \end{align*}
>
> The intuition behind each term is as follows:
>
> - **(A) Attraction** encourages alignment between $\hat{\mathbf{z}}\_{0|T}^{1}$ and its reference $\hat{\mathbf{z}}\_{0|T}^{1,\text{ref}}$, preserving semantic fidelity.
>
> - **(B) Repulsion** pushes $\hat{\mathbf{z}}\_{0|T}^{1}$ away from the other samples in the batch (here, $\hat{\mathbf{z}}\_{0|T}^{2}$), promoting diversity.

---

> ### Author Response · Authors · 2025-11-20
> **Official Comment by Authors (2)**
>
> > **W1-2.** The exact role of gamma in the contrastive loss numerator/denominator, and how gradients are stopped or propagated through each component, could be clarified further. For readers less familiar with contrastive objectives applied to image generation, a stepwise, worked example would reduce ambiguity.
>
> **2. The Role of Gamma ($\gamma$)**
>
> $\gamma$ is a key control parameter introduced in CNO to regulate how strongly each sample adheres to its reference during optimization. In particular, it exclusively scales the attraction term, providing a direct and interpretable mechanism for adjusting the fidelity-diversity trade-off. To see its role more concretely, let us recall the decomposed loss function $\mathcal{L}\_1$:
> \begin{align*}
> \mathcal{L}\_1 = \underbrace{- \frac{\text{sim}(\hat{\mathbf{z}}\_{0|T}^{1}, \hat{\mathbf{z}}\_{0|T}^{1,\text{ref}})}{\gamma\tau}}\_{\text{Attraction}} + \underbrace{\log \left( \exp(\text{sim}(\hat{\mathbf{z}}\_{0|T}^{1}, \hat{\mathbf{z}}\_{0|T}^{1})/\tau)  + \exp(\text{sim}(\hat{\mathbf{z}}\_{0|T}^{1}, \hat{\mathbf{z}}\_{0|T}^{2})/\tau) \right)}\_{\text{Repulsion}}
> \end{align*}
>
> We observe that $\gamma$ is incorporated only in the attraction term, directly scaling its magnitude and thereby controlling the strength of the attraction force. Theoretically, as shown in Proposition 2, $\gamma$ modulates the strength of the positive (attraction) pairs by the factor $1/\gamma$:
> \begin{align*}
> \mathcal{L}^{\gamma}\_{\text{CNO}} \geq - {  \frac{1}{\gamma} }I(Z; Z\_{\text{pos}}) + I(Z;Z\_{\text{neg}})+ \log(B-1),
> \end{align*}
>
>
> Here, we see that decreasing $\gamma$ below $1.0$ would boost the impact of positive terms (e.g., the attraction term), while increasing it over $1.0$ would diminish their effect.
>
> This formulation makes the effect of $\gamma$ explicit:
>
> - **Decreasing $\gamma$ below 1** amplifies the influence of the positive terms (i.e., the attraction), promoting higher fidelity to the reference.
>
>
> - **Increasing $\gamma$ above 1** reduces their contribution, making the optimization more repulsion-driven and promoting greater diversity.
>
> A key practical role of $\gamma$ is to stabilize the fidelity-diversity balance under varying batch sizes $B$. To illustrate this, consider a slightly larger batch size $B = 3$. In this setting, the decomposed loss for $\mathbf{z}\_T^{1}$ becomes:
>
> \begin{align*}
> \mathcal{L}\_1 = \underbrace{- \frac{\text{sim}(\hat{\mathbf{z}}\_{0|T}^{1}, \hat{\mathbf{z}}\_{0|T}^{1,\text{ref}})}{\gamma\tau}}\_{\text{Attraction}} + \underbrace{\log \left( \exp(\text{sim}(\hat{\mathbf{z}}\_{0|T}^{1}, \hat{\mathbf{z}}\_{0|T}^{1})/\tau)  + \exp(\text{sim}(\hat{\mathbf{z}}\_{0|T}^{1}, \hat{\mathbf{z}}\_{0|T}^{2})/\tau) + \exp(\text{sim}(\hat{\mathbf{z}}\_{0|T}^{1}, \hat{\mathbf{z}}\_{0|T}^{3})/\tau) \right)   }\_{\text{Repulsion}},
> \end{align*}
>
> where $\mathcal{L}\_{\text{CNO}}^{\gamma} = \mathcal{L}\_1 + \mathcal{L}\_2 + \mathcal{L}\_3$.  Compared to the case of $B=2$, the repulsion term now includes an additional negative pair involving $\hat{\mathbf{z}}\_{0|T}^{3}$, producing an even stronger repulsive force.
>
> As this example illustrates, when the batch size $B$ is large, the attraction induced by a single reference point (e.g., $\hat{\mathbf{z}}\_{0|T}^{1,\text{ref}}$) can be overshadowed by the $(B-1)$ repulsion terms contributed by the other samples $\lbrace \hat{\mathbf{z}}\_{0|T}^{j,\text{ref}} \rbrace_{j=2}^B$. In such cases, choosing $\gamma < 1$ amplifies the attraction term ($1/\gamma > 1$), yielding a more balanced interplay between fidelity and diversity. Conversely, when $B$ is small, the repulsion effect is much weaker, and overly strong attraction may cause over-alignment with the anchor. Selecting $\gamma$ closer to $1$ (or slightly above) mitigates this issue by preventing over-concentration and preserving sufficient diversity. For completeness, we also provide a closed-form expression for choosing $\gamma$ that offers stable performance across different batch sizes (Eq. (9)):
>
> \begin{align*}
> \gamma = (\tau \text{log} (B-1) + 1)^{-1}.
> \end{align*}

---

> ### Author Response · Authors · 2025-11-20
> **Official Comment by Authors (3)**
>
> > **W1-2.** The exact role of gamma in the contrastive loss numerator/denominator, and how gradients are stopped or propagated through each component, could be clarified further. For readers less familiar with contrastive objectives applied to image generation, a stepwise, worked example would reduce ambiguity.
>
> **3. Gradient Flow and the Stop-Gradient Strategy**
>
> To understand how the optimization updates the latent noise, let us examine the gradient flow using the $B=2$ example. Recall that the loss for the first sample, $\mathcal{L}\_1$, is defined as:$$\mathcal{L}\_1 = -\log \frac{\exp(\text{sim}(\hat{\mathbf{z}}\_{0|T}^{1}, \hat{\mathbf{z}}\_{0|T}^{1,\text{ref}})/(\gamma\tau))}{\exp(\text{sim}(\hat{\mathbf{z}}\_{0|T}^{1}, \hat{\mathbf{z}}\_{0|T}^{1})/\tau) + \exp(\text{sim}(\hat{\mathbf{z}}\_{0|T}^{1}, \hat{\mathbf{z}}\_{0|T}^{2})/\tau)}$$
>
> When computing the gradient with respect to the input noise $\mathbf{z}\_T^{1}$:
>
> **Active gradient flow**: Gradients backpropagate through $\hat{\mathbf{z}}\_{0|T}^{1}$, which appears in both the numerator (attraction) and denominator (repulsion). This drives $\hat{\mathbf{z}}\_{0|T}^{1}$ to move closer to the anchor and further from $\hat{\mathbf{z}}\_{0|T}^{2}$.
>
> **No gradient flow to anchor term**: Crucially, the anchor $\hat{\mathbf{z}}\_{0|T}^{1,\text{ref}}$ is treated as a fixed constant. Therefore, no gradients flow through the reference term. This ensures the anchor remains a stable guidepost throughout the optimization.
>
> However, a challenge arises when propagating this gradient from the denoised estimate $\hat{\mathbf{z}}\_{0|T}$ back to the input noise $\mathbf{z}\_T$. By the chain rule, the gradient is:
>
> $$\nabla\_{\mathbf{z}\_T} \mathcal{L} = \left( \frac{\partial \hat{\mathbf{z}}\_{0|T}}{\partial \mathbf{z}\_T} \right)^T \nabla\_{\hat{\mathbf{z}}\_{0|T}} \mathcal{L}$$
>
> Recalling Tweedie's formula $\hat{\mathbf{z}}\_{0|T} = \frac{1}{\sqrt{\bar{\alpha}\_T}} (\mathbf{z}\_T - \sqrt{1-\bar{\alpha}\_T}\boldsymbol{\epsilon}\_\theta(\mathbf{z}\_T))$, the exact derivative term $\frac{\partial \hat{\mathbf{z}}\_{0|T}}{\partial \mathbf{z}\_T}$ is given by:
>
> $$\frac{\partial \hat{\mathbf{z}}\_{0|T}}{\partial \mathbf{z}\_T} = \frac{1}{\sqrt{\bar{\alpha}\_T}} \left( \mathbf{I} - \sqrt{1-\bar{\alpha}\_T} \frac{\partial \boldsymbol{\epsilon}\_\theta(\mathbf{z}\_T)}{\partial \mathbf{z}\_T} \right)$$
>
> Calculating the term $\frac{\partial \boldsymbol{\epsilon}\_\theta}{\partial \mathbf{z}\_T}$ requires backpropagating through the entire U-Net, which is computationally expensive and prone to unstable high-frequency gradients. To address this, we apply a stopgrad operation to the diffusion model output $\boldsymbol{\epsilon}\_\theta$. This effectively sets the Jacobian of the denoiser to zero ($\frac{\partial \boldsymbol{\epsilon}\_\theta}{\partial \mathbf{z}\_T} \approx 0$) during the backward pass. Consequently, the gradient update simplifies to a scaled version of the identity matrix:
>
> $$\nabla\_{\mathbf{z}\_T} \mathcal{L} \approx \frac{1}{\sqrt{\bar{\alpha}\_T}} \cdot \mathbf{I} \cdot \nabla\_{\hat{\mathbf{z}}\_{0|T}} \mathcal{L}$$
>
> This simplification implies that we update the noise $\mathbf{z}\_T$ directly in the direction that optimizes the contrastive objective in the semantic data space. This strategy mirrors the effective Jacobian approximation successfully employed in methods like Score Distillation Sampling (SDS) [1] and NoiseRefine [2], where the computationally expensive U-Net Jacobian is omitted to ensure stable and efficient guidance.

---

> ### Author Response · Authors · 2025-11-20
> **Official Comment by Authors (4)**
>
> > **W2.** Section 5.1 references an ablation in Figure 4 testing w = (4, 8, 16, 64), yet the impact on specific image types, prompt complexity, or semantic regions is not deeply probed. There is no discussion of whether aggressive downsampling disproportionately harms long-tail or compositional prompts.
>
> Thank you for your insightful observation regarding the downsampling ablation. We have expanded our analysis to more carefully examine how different values of $w$ affect various prompt types. Specifically, we compared the performance of $w = 64$ and $w = 1$ on two representative prompt categories: simple captions (evaluated using GenEval [3]) and complex/compositional captions (evaluated using T2I-CompBench [4]). The quantitative results are shown in the table below:
>
>
> | **Dataset**     | **Setting** | CLIPScore ↑ | PickScore ↑ | ImReward ↑ | MSS ↓ | Vendi Score ↑ |
> |:----------------|:-----------:|:-----------:|:------------:|:-----------:|:------:|:--------------:|
> | **GenEval**     | **$w = 64$**  | 30.7080     | **21.3857**      | **0.0585**      | 0.1017 | **4.8376**         |
> |                 | **$w = 1$**   | **30.8394**     | 21.3320      | 0.0083      | **0.1016** | 4.8374         |
> | **T2I-CompBen** | **$w = 64$**  | 30.8639     | **20.0979**      | **-0.1697**     | 0.1218 | **4.7931**         |
> |                 | **$w = 1$**   | **30.9414**     | 20.0856      | -0.2335     | **0.1208** | 4.7879         |
>
> We observe that for both prompt types, using $w = 1$ yields degraded performance compared to $w = 64$, consistent with the trends reported in our main paper (e.g., Figure 4). Notably, the degradation is more pronounced on T2I-CompBench, where prompts exhibit higher compositional complexity. This suggests that aggressive downsampling may indeed disproportionately harm compositional or long-tail prompts, aligning with the reviewer’s expectation. In our revision, we have incorporated a detailed discussion on this point. See Section C.4 for details.
>
> ---
>
> > **W3.** While automated metrics (Vendi, CLIPScore, etc.) are robust and reflect recent trends, there is a notable absence of human evaluation for subjective diversity, creativity, or aesthetic ranking beyond PickScore. As text-to-image generation is ultimately a user-facing task, such a study would strongly solidify the real-world value of the method.
>
> We agree that human evaluation is important for assessing subjective aspects of diversity and aesthetics. To address this point, we have added a human evaluation study in our revision. In this study, participants were presented with paired outputs, with one image produced by CNO and the other by an existing diversity-oriented method. For each pair, they were asked to choose the preferred image under two separate criteria: (i) perceived diversity and creativity, and (ii) aesthetic appeal and text-image consistency. The aggregated results are shown in the table below:
>
> |                      | Alignment| & Quality (\%) \|           | Diversity | & Creativity (\%)           |
> |----------------------|:---:|:---:|:---:|:---:|
> |                      | Baseline            | **CNO**    | Baseline                 | **CNO**    |
> | **vs. CADS**         | 21.29               | **78.71**  | 16.45                    | **83.55**  |
> | **vs. PG**           | 41.29               | **58.71**  | 21.94                    | **78.06**  |
>
> Note that participants consistently preferred CNO over the baselines on both criteria, indicating that our method provides improvements that are also meaningful from a human perceptual perspective. These findings are aligned with the trends observed in automated metrics (see Table 1). Additional details on the user study are provided in Section D.2 of our revised manuscript.

---

> ### Author Response · Authors · 2025-11-20
> **Official Comment by Authors (5)**
>
> > **Q1-1.** Could the authors provide additional quantitative and qualitative results on how extreme values of gamma  (especially <<1  and >>1) impact the fidelity/diversity trade-off?
>
> As per your suggestion, we investigated the effect of using extreme values of $\gamma$ on the fidelity-diversity trade-off. Specifically, we evaluated $\gamma = 0.01$ and $\gamma = 100.0$, and compared their diversity and fidelity metrics against DDIM and our nominal setting ($\gamma = 1.0$). The detailed results are summarized in the table below:
>
> |                    | CLIPScore ↑ | PickScore ↑ | ImReward ↑ | MSS ↓  | Vendi Score ↑ |
> |:--------------------------------|:-----------:|:------------:|:-----------:|:------:|:--------------:|
> | **DDIM**                        | **31.3940** | 21.3937      | **0.0890**  | 0.1485 | 4.7413         |
> | **CNO ($\gamma = 0.01$)**           | 31.3934     | **21.3945**  | 0.0888      | 0.1485 | 4.7412         |
> | **CNO ($\gamma = 1.0$)**            | 31.3424     | 21.2907      | 0.0360      | **0.1276** | **4.7945** |
> | **CNO ($\gamma = 100.0$)**          | 31.3764     | 21.2696      | 0.0185      | 0.1279 | 4.7933         |
>
> We observe that $\gamma = 0.01$ produces results that are nearly identical to DDIM. This behavior is expected because the attraction toward the fixed Tweedie estimate $\hat{\mathbf{z}}_{0|T}^{i, \text{ref}}$ becomes overwhelmingly strong, effectively suppressing the repulsion term that promotes diversity. In contrast, $\gamma = 100.0$ increases the strength of the repulsive component, leading to a modest improvement in diversity. However, this comes at the cost of degraded fidelity, as the attraction term becomes too weak to enforce adequate quality regularization. As a result, the overall fidelity-diversity balance becomes worse than that of our nominal setting $\gamma = 1.0$. These findings, along with further discussion, are now included in our revised manuscript. See Section C.2 for more details.
>
> ---
>
> > **Q1-2.** Are there prompts or settings where the method consistently fails or collapses into poor outputs?
>
> In fact, our method does exhibit limitations, particularly in prompts requiring strong compositional grounding. In such cases, CNO may compromise text-image alignment, which typically appears as missing or inaccurately rendered objects or relations in complex multi-entity prompts; see Figure 9 in our revision.
>
> In the example, DDIM captures all components of the prompt, such as the dog, the yellow frisbee, and the holding relation. In contrast, diversity-enhancing approaches such as CADS and PG fail to preserve the relational structure, especially the correct depiction of holding. While our method at the nominal setting ($\gamma=1.0$) also faces this challenge due to the inherent fidelity-diversity trade-off, it offers a flexible solution through the adjustable attraction coefficient.
>
> As demonstrated in Figure 9(e), by lowering the attraction coefficient (e.g., $\gamma < 1.0$), CNO can intensify the anchoring force toward the initial Tweedie estimate. This effectively recovers the correct compositional structure (the holding relation) that other diversity methods miss, while still offering diversity benefits over DDIM. This confirms that $\gamma$ serves as an effective fidelity controller, allowing users to flexibly resolve failure cases that remain fixed limitations in other baselines. We have added a detailed discussion of this controllable trade-off in our revised manuscript; see Section C.8 for details.

---

> ### Author Response · Authors · 2025-11-20
> **Official Comment by Authors (6)**
>
> > **Q2-1.** What are the empirical impacts and best practices for choosing the batch size B in real-world usage?
>
> Empirically, we found that the best practice for selecting $B$ is to set it equal to the number of images one intends to generate per prompt. For instance, if a user wishes to generate 5 images from a single prompt, choosing $B = 5$ leads to a strong fidelity-diversity balance, while using smaller values tends to yield weaker performance.
>
> This rule-of-thumb arises naturally from how CNO operates. The maximum meaningful choice of $B$ is the number of images per prompt (NIPP for short), since setting $B$ larger would mix samples from different prompts, which are already sufficiently diverse due to prompt incoherency. Increasing $B$ toward this upper bound allows the repulsion term to push noise samples into increasingly diverse directions that are not represented in the current batch. As a result, larger $B$ encourages exploration of a broader span of latent modes, improving diversity with only marginal quality degradation. The table below illustrates this behavior for the case where the NIPP is set to 5.
>
> |    | CLIPScore ↑ | PickScore ↑ | ImReward ↑ | MSS ↓   | Vendi Score ↑ |
> |:------------------------|:-----------:|:------------:|:-----------:|:-------:|:--------------:|
> | **DDIM**                | **31.3940**     | **21.3937**      | **0.0890**  | 0.1485  | 4.7413         |
> | **CNO** ($B=3$)         | 31.3298     | 21.2640      | 0.0238      | **0.1267** | **4.7949**    |
> | **CNO** ($B=5$)         | 31.3424     | 21.2907      | 0.0360      | 0.1276  | 4.7945         |
>
> We observe that $B = 5$ (equal to the NIPP) achieves the best trade-off. Nonetheless, smaller choices such as $B = 3$ still provide meaningful diversity gains, indicating that CNO is not overly sensitive to $B$. In practice, the recommended guideline is to select the largest $B$ allowed by memory, up to the NIPP.
>
> As a complementary analysis, we also report the computational cost associated with different $B$ values; see the table below.
>
> |   |        **$B = 3$**        |        |        **$B = 5$**        |        |
> |---------|:------------------------:|:------:|:------------------------:|:------:|
> |           | **Memory (MiB) ↓** | **Time (sec/batch) ↓** | **Memory (MiB) ↓** | **Time (sec/batch) ↓** |
> | **DDIM**  | 7694         | **8.091**        | 10978        | **12.778**       |
> | **CNO**   | **7672**         | 8.809        | **10828**        | 13.427       |
> | **DiversityPrompt** | 13070 | 15.063 | 19006 | 21.631 |
>
> While increasing $B$ naturally increases memory usage and runtime, the overhead for CNO remains modest compared to existing diversity-enhancing approaches such as DiversityPrompt, which incurs nearly double the inference time and memory consumption. We have incorporated these findings and their accompanying discussion into our revised manuscript; see Section C.7 for details.
>
> ---
>
> > **Q2-2.** Is there a threshold beyond which diversity improvements stagnate or degrade?
>
> Based on our experiments (e.g., the table above), we observe that diversity continues to improve as $B$ approaches the number of images per prompt, and we rarely see signs of stagnation or degradation within this range.

---

> ### Author Response · Authors · 2025-11-20
> **Official Comment by Authors (7)**
>
> > **Q3-1.** Is the optimization of the initial noise vectors consistently stable across different diffusion model architectures?
>
> We found that CNO remains stable and consistently improves diversity across a wide range of diffusion architectures and model families. In particular, our method applies to both U-Net-based and MMDiT-based backbones, and even to accelerated few-step samplers. As shown in Table 1 of the manuscript, CNO yields consistent gains in SD1.5, SDXL, and SD3, covering both architectural types.
>
> To further support the applicability of our approach under few-step frameworks, we additionally evaluated CNO on FLUX-1-Schnell (4-step) and SDXL-Lightning (4-step), which are structurally distinct rectified-flow and distilled models where diversity degradation is typically more severe. The results are summarized below:
>
>
> | FLUX-1 | CLIPScore ↑ | PickScore ↑ | ImReward ↑ | MSS ↓ | Vendi Score ↑ |
> |:-------|:-----------:|:------------:|:-----------:|:------:|:---------------:|
> | **FM-ODE** | **32.1012** | **22.7411** | **1.0499** | 0.3012 | 2.7220 |
> | **CADS** | 31.7153 | 22.5431 | 0.8622 | 0.2287 | 2.8250 |
> | **CNO** | 32.0664 | 22.6137 | 1.0070 | **0.2231** | **2.8316** |
>
>
> | SDXL-LT | CLIPScore ↑ | PickScore ↑ | ImReward ↑ | MSS ↓ | Vendi Score ↑ |
> |---|:---:|:---:|:---:|:---:|:---:|
> | **DDIM** | **31.5536** | **22.6598** | **0.7231** | 0.2865 | 2.7470 |
> | **CADS** | 31.4425 | 22.5767 | 0.6876 | 0.2674 | 2.7749 |
> | **CNO** | 31.4474 | 22.5659 | 0.6740 | **0.2289** | **2.8258** |
>
> We observe consistent improvements across these highly compressed few-step settings, while previous approaches like CADS give marginal improvements. This demonstrates that CNO is broadly applicable and robust across different diffusion architectures and inference regimes. These results, along with additional discussion, have been incorporated into our revised manuscript; see Section D.1 for details.
>
> As an additional remark, we provide some intuition on why CNO offers robust diversity improvements even in few-step settings, where existing diversity samplers often struggle. CNO performs a one-shot optimization of the initial noise guided by Tweedie estimates $\hat{\mathbf{z}}_{0|T}^i$, and the only requirement for this step to be effective is that the Tweedie predictions preserve meaningful structural cues about the eventual samples $\mathbf{z}_0^i$. Interestingly, we found that accelerated few-step models tend to produce clearer and more deterministic Tweedie estimates than standard many-step samplers (see Figure 10), which makes them particularly compatible with our approach.
>
> In contrast, existing diversity-oriented samplers (such as CADS) rely on iterative interventions during sampling. When the sampler has only a few steps, these interventions have very limited opportunity to influence the trajectory, leading to only marginal gains. This explains why CNO maintains strong performance in these regimes, whereas intervention-based methods typically do not.
>
> ---
>
> > **Q3-2.** Are there architectures/settings for which CNO performs suboptimally?
>
> As mentioned in Q1-2, we did find some failure cases with particularly challenging prompts, such as overly complex or semantically overloaded text. In such cases, CNO occasionally struggles to maintain sample quality. These examples are now included in Figure 9 of the revised manuscript.
>
> ---
>
> **References**
>
> [1] DreamFusion: Text-to-3D using 2D Diffusion, ICLR 2023.
>
> [2] A Noise is Worth Diffusion Guidance, Arxiv 2024.
>
> [3] GenEval: An Object-Focused Framework for Evaluating Text-to-Image Alignment, NeurIPS 2023.
>
> [4] T2I-CompBench: A Comprehensive Benchmark for Open-world Compositional Text-to-image Generation, NeurIPS 2023.

---

### Official Review · Reviewer_qqYa · 2025-10-28

**Soundness:** 3
**Presentation:** 3
**Contribution:** 3
**Rating:** 6
**Confidence:** 5

**Summary:**

The paper presents Contrastive Noise Optimization (CNO), a training-free diversity-boosting wrapper for text-to-image diffusion models. By marrying Tweedie-space contrastive repulsion with anchor-guided fidelity control, CNO produces markedly varied outputs in a single, lightweight pre-processing stage. Comprehensive evaluations on SD 1.5, SDXL and SD3 reveal consistent gains in Vendi score with negligible CLIPScore degradation, surpassing the latest zero-shot samplers in MSS and human-preference metrics and establishing a new Pareto frontier for quality-diversity trade-offs.

**Strengths:**

1.	Diversity is an important and interesting topic. Many models tend to overlook diversity issues during pretraining.
2.	With complex prompts like “A cow sits in a truck with hay barrels in it,” other methods either fail or repeat elements, but our method produces diverse and semantically accurate images.

**Weaknesses:**

1. The proposed method improves generation diversity by optimizing the initial latent, which is sampled from a Gaussian distribution. As the initial latent is determined by this distribution, adjusting the variance of the Gaussian may also affect diversity. It would be interesting to discuss whether such a change could further enhance diversity.
2. It might be worth exploring whether the proposed method can further improve the diversity of the few-step distilled models.
3. A user study could help verify, through human evaluation, whether the method achieves higher generative diversity than the baseline.
4. It would be helpful if the authors could report the additional computational resources required as B increases.

**Questions:**

The proposed method shows strong qualitative and quantitative performance across multiple models (SD1.5, SDXL, SD3). However, as highlighted in Diffusion2GAN and Loopfree, few-step models often exhibit more severe diversity degradation. It would be valuable to further analyze whether the proposed method remains effective under such few-step models.

Diffuison2GAN: Distilling Diffusion Models into  Conditional GANs \
Loopfree: One-Way Ticket : Time-Independent Unified Encoder for Distilling Text-to-Image Diffusion Models

---

> ### Author Response · Authors · 2025-11-20
> **Official Comment by Authors (1)**
>
> We thank Reviewer `qqYa` for the insightful comments and constructive suggestions. Below, we provide point-by-point responses addressing each concern.
>
> ---
> > **W1.** The proposed method improves generation diversity by optimizing the initial latent, which is sampled from a Gaussian distribution. As the initial latent is determined by this distribution, adjusting the variance of the Gaussian may also affect diversity. It would be interesting to discuss whether such a change could further enhance diversity.
>
> As per your suggestion, we explored whether controlling the variance of the Gaussian prior distribution can further enhance sample diversity. Specifically, we introduced a multiplicative factor $\tau$ into the prior and sampled the initial latent as $\mathbf{z}_T \sim {\cal N}({\boldsymbol 0}, \tau^2 {\boldsymbol I})$. See the table below for the quantitative results:
>
> |             | CLIPScore ↑ | PickScore ↑ | ImReward ↑ | MSS ↓   | Vendi Score ↑ |
> |---------------------|:-----------:|:------------:|:-----------:|:-------:|:--------------:|
> | **$\tau = 1.0$ (DDIM)**   | **31.5041** | 21.4358 | 0.1324 | 0.1620 | 4.7029 |
> | **$\tau = 1.01$**         | 31.4950 | **21.4381** | 0.1597 | 0.1608 | 4.7060 |
> | **$\tau = 1.025$**        | 31.4668 | 21.4313 | 0.1814 | 0.1599 | 4.7090 |
> | **$\tau = 1.05$**         | 31.4469 | 21.3237 | **0.2058** | 0.1621 | 4.7058 |
> | **$\tau = 1.1$**          | 31.4757 | 20.8230 | 0.0817 | 0.1811 | 4.6539 |
> | **$\tau = 1.15$**         | 30.0744 | 19.6513 | -0.6910 | 0.2114 | 4.5561 |
> | **CNO (ours)**                   | 31.3424 | 21.2907 | 0.0360 | **0.1276** | **4.7945** |
>
>
>
> We observe that slightly increasing $\tau$ above 1.0 yields only marginal improvements in diversity metrics, whereas larger values result in a clear degradation of overall generation quality without offering additional diversity benefits. This suggests that naively enlarging the variance of the prior is not an effective strategy for improving diversity. Interestingly, this trend is consistent with recent findings [1], which report that inflating the variance in non-stochastic ODE-based T2I samplers can substantially deteriorate sample quality. We have incorporated these findings into our revised manuscript, along with a more detailed discussion in Section C.5.

---

> ### Author Response · Authors · 2025-11-20
> **Official Comment by Authors (2)**
>
> > **W2 / Q1.** It might be worth exploring whether the proposed method can further improve the diversity of the few-step distilled models. … However, as highlighted in Diffusion2GAN and Loopfree, few-step models often exhibit more severe diversity degradation. It would be valuable to further analyze whether the proposed method remains effective under such few-step models.
>
>
> We appreciate the reviewer’s suggestion to examine whether our method remains effective for few-step distilled models, which are indeed known to suffer from more severe diversity degradation. To assess this, we conducted additional experiments on SDXL-Lightning (4-step), a highly distilled variant of SDXL. The results are summarized below:
>
> |  | CLIPScore ↑ | PickScore ↑ | ImReward ↑ | MSS ↓ | Vendi Score ↑ |
> |---|:---:|:---:|:---:|:---:|:---:|
> | **DDIM** | **31.5536** | **22.6598** | **0.7231** | 0.2865 | 2.7470 |
> | **CADS** | 31.4425 | 22.5767 | 0.6876 | 0.2674 | 2.7749 |
> | **CNO (ours)** | 31.4474 | 22.5659 | 0.6740 | **0.2289** | **2.8258** |
>
> We observe that CNO provides consistent improvements even under this extremely distilled 4-step setting, demonstrating that our method remains effective for both standard and distilled diffusion frameworks. In contrast, existing diversity-oriented samplers such as CADS show limited gains in this regime. We believe this is due to their reliance on iterative interventions during sampling, which offers little benefit when the sampler has very few steps and thus very few opportunities to intervene.
>
>
> Notably, CNO avoids this limitation by performing a one-shot optimization of the initial noise based upon Tweedie estimates $\hat{\mathbf{z}}_{0|T}^i$. The only requirement for this step to be effective is that the Tweedie predictions retain meaningful structural information about the eventual samples $\mathbf{z}_0^i$. Interestingly, we found that accelerated few-step models often produce clearer and more deterministic Tweedie estimates than many-step samplers (see Figure 10), which explains why CNO remains particularly effective in these settings.
>
>
> As additional evidence that CNO performs well across various types of few-step models, we also evaluated our method on **FLUX-1-Schnell**, a structurally distinct rectified-flow model that employs aggressively accelerated sampling. The results are shown below:
>
>
> |  | CLIPScore ↑ | PickScore ↑ | ImReward ↑ | MSS ↓ | Vendi Score ↑ |
> |:-------|:-----------:|:------------:|:-----------:|:------:|:---------------:|
> | **FM-ODE** | **32.1012** | **22.7411** | **1.0499** | 0.3012 | 2.7220 |
> | **CADS** | 31.7153 | 22.5431 | 0.8622 | 0.2287 | 2.8250 |
> | **CNO (ours)** | 32.0664 | 22.6137 | 1.0070 | **0.2231** | **2.8316** |
>
>
> We again observe consistent diversity improvements over the baselines, indicating that our approach is broadly applicable across different accelerated sampling approaches. These findings and their implications have been incorporated into our revised manuscript; see Section D.1 for details.

---

> ### Author Response · Authors · 2025-11-20
> **Official Comment by Authors (3)**
>
> > **W3.** A user study could help verify, through human evaluation, whether the method achieves higher generative diversity than the baseline.
>
> We have now incorporated a human evaluation in our revision, where we compare our method against existing diversity approaches based on human preference. In this study, participants were shown paired outputs consisting of one image from CNO and one from a baseline method. To ensure fairness, the presentation order of the two methods was randomized for every comparison. For each pair, evaluators were asked to select the preferred output along two independent criteria: (i) diversity and creativity, and (ii) aesthetic quality and text-image alignment. The results are summarized in the table below:
>
> |                      | Alignment| & Quality (\%) \|           | Diversity | & Creativity (\%)           |
> |----------------------|:---:|:---:|:---:|:---:|
> |                      | Baseline            | **CNO (ours)**    | Baseline                 | **CNO (ours)**    |
> | **vs. CADS**         | 21.29               | **78.71**  | 16.45                    | **83.55**  |
> | **vs. PG**           | 41.29               | **58.71**  | 21.94                    | **78.06**  |
>
> Observe that our method is consistently preferred over the baselines across both criteria, indicating that CNO produces samples that humans perceive as more diverse and of higher visual and semantic quality. Notably, this trend aligns well with the automatic evaluation metrics (see Table 1). Further details on the human evaluation are provided in Section D.2 of our revision.
>
> ---
>
> > **W4.** It would be helpful if the authors could report the additional computational resources required as B increases.
>
> To reflect your comment, we evaluated how the computational cost of our method scales with the batch size $B$. Specifically, we measured the peak memory consumption and wall-clock sampling time per batch for $B = 3$ and $B = 5$. The results are summarized below:
>
> |   |        **$B = 3$**        |        |        **$B = 5$**        |        |
> |---------|:------------------------:|:------:|:------------------------:|:------:|
> |           | **Memory (MiB) ↓** | **Time (sec/batch) ↓** | **Memory (MiB) ↓** | **Time (sec/batch) ↓** |
> | **DDIM**  | 7694         | **8.091**        | 10978        | **12.778**       |
> | **CNO**   | **7672**         | 8.809        | **10828**        | 13.427       |
> | **DiversityPrompt** | 13070 | 15.063 | 19006 | 21.631 |
>
>
> We see that increasing $B$ naturally leads to higher computational requirements for our approach. Nonetheless, additional computational overhead of CNO across various $B$ is relatively marginal compared to existing baselines (like DiversityPrompt), demonstrating the practical feasibility of our method. We highlight that this benefit is due to our perspective of one-shot initial noise selection before generation, which is inherently efficient than previous diversity-focused samplers that rely upon iterative inference-time interventions. We have included this point in our revision; see Section C.7 for details.
>
> As an additional remark, we also found that CNO yields meaningful diversity gains even with relatively small batch sizes; see the table below for instance:
>
>
> |    | CLIPScore ↑ | PickScore ↑ | ImReward ↑ | MSS ↓   | Vendi Score ↑ |
> |:------------------------|:-----------:|:------------:|:-----------:|:-------:|:--------------:|
> | **DDIM**                | **31.3940**     | **21.3937**      | **0.0890**  | 0.1485  | 4.7413         |
> | **CNO** ($B=3$)         | 31.3298     | 21.2640      | 0.0238      | **0.1267** | **4.7949**    |
> | **CNO** ($B=5$)         | 31.3424     | 21.2907      | 0.0360      | 0.1276  | 4.7945         |
>
>
> ---
>
> **References**
>
> [1] Boost-and-Skip: A Simple Guidance-Free Diffusion for Minority Generation, ICML 2025.

---

### Official Review · Reviewer_XTmn · 2025-11-06

**Soundness:** 2
**Presentation:** 2
**Contribution:** 2
**Rating:** 4
**Confidence:** 5

**Summary:**

This paper addresses the persistent issue of limited diversity in text-to-image (T2I) diffusion models under strong text guidance. Instead of prior approaches that tinker with intermediate latents or guide text embeddings during or across inference steps, the authors introduce Contrastive Noise Optimization (CNO): a lightweight, pre-processing method that optimizes the initial noise latents before sampling. By defining a contrastive InfoNCE loss in the Tweedie denoised latent space—incorporating both attractive (anchoring) and repulsive (diversifying) forces—CNO aims to produce batches of noise that yield diverse yet faithful image outputs. The method is anchored in theoretical reformulation of InfoNCE to balance fidelity and diversity, is computationally efficient, and demonstrates robust gains across several quantitative and qualitative metrics on Stable Diffusion variants and benchmarks.

**Strengths:**

The paper is technically sound, mathematically rigorous in its core claims, and offers strong theoretical motivation for adopting a contrastive loss in the denoised Tweedie latent space. The derivations for the mutual information bounds (Propositions 1 & 2) and the use of a regularization parameter ($\gamma$) are helpful and clear (see Sections 4.2–4.3 and Appendix A).

Extensive experiments (see Table 1 and Figure 3) across SD1.5, SDXL, and SD3 show the method consistently outperforms recent strong baselines like CADS, PG, and DiversityPrompt, especially on diversity metrics (Vendi Score, MSS) while retaining high-quality image generation (PickScore, ImageReward).

**Weaknesses:**

The loss function as written in Section 4.1/Algorithm 1 and its implementation in Algorithm 2 (for KL regularization) are slightly inconsistent in the summation indices and argument ordering, which could lead to confusion for others seeking to re-implement the approach.

The diversity-boosting methods compared against cover PG, CADS, and DiversityPrompt; however, some recent methods involving multi-concept fusion, 3D-aware T2I generation, and counterfactual interventions (cf. TweedieMix (Kwon & Ye 2025), DiffSplat (Lin et al. 2025), CoT-lized Diffusion (Liu et al. 2025), and Pan & Bareinboim (2025)) are not considered or even discussed. These methods, while not identical, are highly relevant to the diversity and compositional generalization challenge, and their exclusion diminishes the scope of empirical claims.

The main conceptual innovation of optimizing the initial noise with a contrastive loss heavily draws from existing InfoNCE/contrastive techniques, which are increasingly used in generative and diffusion-based frameworks. The extension to the Tweedie latent space is a clever adaptation, though it may be seen as an incremental, albeit meaningful, refinement of existing paradigms (see Guo et al. 2024, Ahn et al. 2024). The contribution would have benefited from a more careful positioning, clarifying the distinction between novelty and adaptation.

This paper provides a practical contribution to advancing diversity in text-to-image generative models.  The empirical evaluation is thorough, with solid quantitative and qualitative evidence supporting the claims of improved Pareto frontiers.

However, the work is somewhat incremental in terms of conceptual novelty, with the main technical move (contrastive optimization of initial noise) being an adaptation of ideas familiar within the generative/representation learning community. There are also missing references to closely related state-of-the-art works on multi-concept and compositional generation, diversity quantification, and alternative diversity control strategies, some of which could serve as strong baselines. Additionally, the exposition could benefit from clarified notation and more critical discussion of potential failure corners.

**Questions:**

The loss function as written in Section 4.1/Algorithm 1 and its implementation in Algorithm 2 (for KL regularization) are slightly inconsistent in the summation indices and argument ordering, which could lead to confusion for others seeking to re-implement the approach.

The diversity-boosting methods compared against cover PG, CADS, and DiversityPrompt; however, some recent methods involving multi-concept fusion, 3D-aware T2I generation, and counterfactual interventions (cf. TweedieMix (Kwon & Ye 2025), DiffSplat (Lin et al. 2025), CoT-lized Diffusion (Liu et al. 2025), and Pan & Bareinboim (2025)) are not considered or even discussed. These methods, while not identical, are highly relevant to the diversity and compositional generalization challenge, and their exclusion diminishes the scope of empirical claims.

The main conceptual innovation of optimizing the initial noise with a contrastive loss heavily draws from existing InfoNCE/contrastive techniques, which are increasingly used in generative and diffusion-based frameworks. The extension to the Tweedie latent space is a clever adaptation, though it may be seen as an incremental, albeit meaningful, refinement of existing paradigms (see Guo et al. 2024, Ahn et al. 2024). The contribution would have benefited from a more careful positioning, clarifying the distinction between novelty and adaptation.

This paper provides a practical contribution to advancing diversity in text-to-image generative models.  The empirical evaluation is thorough, with solid quantitative and qualitative evidence supporting the claims of improved Pareto frontiers.

However, the work is somewhat incremental in terms of conceptual novelty, with the main technical move (contrastive optimization of initial noise) being an adaptation of ideas familiar within the generative/representation learning community. There are also missing references to closely related state-of-the-art works on multi-concept and compositional generation, diversity quantification, and alternative diversity control strategies, some of which could serve as strong baselines. Additionally, the exposition could benefit from clarified notation and more critical discussion of potential failure corners.

---

> ### Author Response · Authors · 2025-11-20
> **Official Comment by Authors (1)**
>
> We sincerely thank Reviewer `XTmn` for the thoughtful comments and constructive feedback. Below, we provide detailed point-by-point responses addressing each of your concerns.
>
> ---
>
> > **W1.** The loss function as written in Section 4.1/Algorithm 1 and its implementation in Algorithm 2 (for KL regularization) are slightly inconsistent in the summation indices and argument ordering, which could lead to confusion for others seeking to re-implement the approach. … Additionally, the exposition could benefit from clarified notation … .
>
> Thank you for pointing this out. We agree that the mismatch in the summation indices and argument ordering could cause confusion for reproduction. In the revised manuscript, we carefully unified all related mathematical notations in Figure 2, Section 4 and updated Algorithms 1 and 2 to ensure full consistency. All corresponding revisions are highlighted in blue.
>
> ---
>
> > **W2.** The diversity-boosting methods compared against cover PG, CADS, and DiversityPrompt; however, some recent methods involving multi-concept fusion, 3D-aware T2I generation, and counterfactual interventions (cf. TweedieMix (Kwon & Ye 2025), DiffSplat (Lin et al. 2025), CoT-lized Diffusion (Liu et al. 2025), and Pan & Bareinboim (2025)) are not considered or even discussed. These methods, while not identical, are highly relevant to the diversity and compositional generalization challenge, and their exclusion diminishes the scope of empirical claims. … There are also missing references to closely related state-of-the-art works on multi-concept and compositional generation, diversity quantification, and alternative diversity control strategies, some of which could serve as strong baselines.
>
> To address this concern, we expanded our empirical comparisons to include additional diversity-related techniques. In particular, we evaluated TweedieMix alongside our approach as well as existing diversity-focused baselines. For this comparison, we constructed compositional captions using the personalized concepts provided in TweedieMix (e.g., dog, cat, mountain) and employed the official pretrained weights from its codebase. See the table below for the results:
>
> |       | CLIPScore ↑ | PickScore ↑ | ImReward ↑ | MSS ↓     | Vendi Score ↑ |
> |-------------|:------------:|:--------------:|:-------------:|:-----------:|:----------------:|
> | **DDIM**       | **35.9353**  | **23.1232**   | **1.4163**    | 0.2319    | 2.8302         |
> | **CADS**        | 35.2270    | 22.7042      | 1.1303      | 0.1860    | 2.8816         |
> | **TweedieMix**  | 29.7348    | 20.1795      | -0.4640     | 0.2112    | 2.8531         |
> | **CNO (ours)**         | 35.3717  | 22.7858      | 1.2632      | **0.1645** | **2.9040**      |
>
> Observe that CNO yields the best trade-off between diversity and fidelity, while TweedieMix demonstrates degraded text-image alignment. This degradation is attributable to the notorious catastrophic forgetting issue prevalent in existing personalization approaches, causing them to struggle when generating high-quality images that align well with general text prompts (i.e., when concepts outside the personalized set are involved); please see Figure 8 for an illustration of this phenomenon. Furthermore, the limited diversity gain observed in TweedieMix stems from its reliance on personalized concept embeddings learned through an additional training stage, which naturally biases the model toward those memorized concepts and consequently restricts its ability to explore diverse variations. Another critical limitation is that TweedieMix requires segmenting Tweedie latents using Text-SAM before fusion, which introduces considerable computational overhead and makes the method harder to scale beyond a small number of learned concepts. As a result, TweedieMix tends to remain close to the pre-learned concept representations and is less flexible for arbitrary prompts.
>
> In contrast, our framework requires no extra training, applies directly to any prompt without relying on pre-defined concepts, and improves diversity in a lightweight one-shot manner. In our revision, we have also added a dedicated discussion on other related approaches, including 3D-aware T2I generation and counterfactual interventions. See Section C.6 for details.

---

> ### Author Response · Authors · 2025-11-20
> **Official Comment by Authors (2)**
>
> > **W3.** The main conceptual innovation of optimizing the initial noise with a contrastive loss heavily draws from existing InfoNCE/contrastive techniques, which are increasingly used in generative and diffusion-based frameworks. The extension to the Tweedie latent space is a clever adaptation, though it may be seen as an incremental, albeit meaningful, refinement of existing paradigms (see Guo et al. 2024, Ahn et al. 2024). The contribution would have benefited from a more careful positioning, clarifying the distinction between novelty and adaptation. … However, the work is somewhat incremental in terms of conceptual novelty, with the main technical move (contrastive optimization of initial noise) being an adaptation of ideas familiar within the generative/representation learning community.
>
> Although CNO is indeed inspired by the structure of the InfoNCE objective, we believe our approach offers substantial novelty both conceptually and technically. We summarize the key reasons below.
>
> ### **1. A clear paradigm shift: from inference-time interventions to initial-noise selection**
>
> CNO reframes diversity enhancement in diffusion models as a problem of initial-noise selection. In fact, prior methods depend on *on-the-fly interventions*, either modifying intermediate latents (e.g., PG, ShieldedDiffusion) or incorporating time-varying text embeddings (e.g., CADS, DiversityPrompt). These approaches require repeated adjustments during sampling and thus suffer from notable limitations, including substantial computational overhead (e.g., DiversityPrompt; see Table 4) and limited gains under accelerated few-step models, where opportunities for intervention are intrinsically scarce.
>
> We highlight that CNO avoids these issues entirely by performing an intelligent one-shot optimization of the initial noise before sampling, introducing only negligible overhead compared to standard samplers (see Table 4). Moreover, our method remains effective even in accelerated few-step models, as demonstrated by the results below on SDXL-Lightning (4-step) and FLUX-1-Schnell (4-step):
>
> | SDXL-LT | CLIPScore ↑ | PickScore ↑ | ImReward ↑ | MSS ↓ | Vendi Score ↑ |
> |---|:---:|:---:|:---:|:---:|:---:|
> | **DDIM** | **31.5536** | **22.6598** | **0.7231** | 0.2865 | 2.7470 |
> | **CADS** | 31.4425 | 22.5767 | 0.6876 | 0.2674 | 2.7749 |
> | **CNO (ours)** | 31.4474 | 22.5659 | 0.6740 | **0.2289** | **2.8258** |
>
> | FLUX-1 | CLIPScore ↑ | PickScore ↑ | ImReward ↑ | MSS ↓ | Vendi Score ↑ |
> |:-------|:-----------:|:------------:|:-----------:|:------:|:---------------:|
> | **FM-ODE** | **32.1012** | **22.7411** | **1.0499** | 0.3012 | 2.7220 |
> | **CADS** | 31.7153 | 22.5431 | 0.8622 | 0.2287 | 2.8250 |
> | **CNO (ours)** | 32.0664 | 22.6137 | 1.0070 | **0.2231** | **2.8316** |
>
> The benefit of CNO remains consistent across these highly compressed models, in stark contrast to existing approaches such as CADS. Given the growing importance of fast, few-step diffusion models in real-world applications, enabling substantial diversity improvements without any inference-time guidance is an important conceptual contribution that meaningfully advances the capabilities of current T2I systems.
>
>
> ### **2. Non-trivial extensions beyond InfoNCE, tailored to diverse T2I generation**
>
> CNO is far from a naive reuse of the standard InfoNCE objective. Instead, it introduces structural modifications essential for diverse T2I generation, most notably the balancing coefficient $\gamma$. As shown in Section 4.2, $\gamma$ enables explicit control over the interplay between attraction and repulsion, which is crucial in diverse generation where maintaining fidelity is as important as promoting diversity. This parameter is key to achieving well-balanced performance across different batch sizes $B$, since the relative strength of attraction and repulsion naturally depends on $B$.
>
> Importantly, $\gamma$ is not introduced as a heuristic choice. Rather, its formulation is supported by both theoretical analysis and empirical evidence (Proposition 2 and Section C.2), along with a closed-form expression for selecting an effective value of $\gamma$ (Eq. 9). We emphasize that our fidelity-control mechanism differs fundamentally from existing noise-optimization frameworks such as InitNO, which typically rely on KL regularization with respect to the Gaussian prior. Although the forms differ, we show empirically that $\gamma$ exhibits effects similar to KL-based regularization (Section C.3), further validating the soundness of our formulation.
>
> As an additional contribution, Proposition 1 clarifies the role of negative pairs in the InfoNCE objective, offering conceptual insight often overlooked in prior work.

---

> ### Author Response · Authors · 2025-11-20
> **Official Comment by Authors (3)**
>
> > **W3.** The main conceptual innovation of optimizing the initial noise with a contrastive loss heavily draws from existing InfoNCE/contrastive techniques, which are increasingly used in generative and diffusion-based frameworks. The extension to the Tweedie latent space is a clever adaptation, though it may be seen as an incremental, albeit meaningful, refinement of existing paradigms (see Guo et al. 2024, Ahn et al. 2024). The contribution would have benefited from a more careful positioning, clarifying the distinction between novelty and adaptation. … However, the work is somewhat incremental in terms of conceptual novelty, with the main technical move (contrastive optimization of initial noise) being an adaptation of ideas familiar within the generative/representation learning community.
>
> ### **3. Practical innovations that significantly improve applicability**
>
> Finally, CNO includes practical design choices that are crucial for making contrastive noise optimization efficient and deployable. Techniques such as the adaptive latent pooling and stop-gradient technique (Section 4.1) dramatically reduce computational overhead while preserving performance. These components are not merely engineering tweaks: they are essential for enabling a scalable optimization framework for diverse T2I generation (e.g., See Table 8).
>
> Taken together, these points show that our method (CNO) is not simply an incremental adaptation of existing contrastive techniques. Rather, it reshapes the paradigm of diversity enhancement, introduces non-trivial extensions to contrastive learning tailored to diversity-oriented diffusion samplers, and provides practical mechanisms needed for real-world deployment. We have updated the manuscript to make these distinctions clearer; please refer to the revised version for the corresponding clarifications (Sections 1 and 4.1).
>
> ---
>
> > **W4.** Additionally, the exposition could benefit from … and more critical discussion of potential failure corners.
>
> We agree that discussing potential failure cases strengthens the exposition. Our method does exhibit limitations, particularly in prompts requiring strong compositional grounding. In such scenarios, CNO (at nominal settings) may compromise text-image alignment, which often manifests as missing or inaccurately rendered objects or relations; see Figure 9 in our revision.
>
> In Figure 9, DDIM successfully captures all components, such as the dog, the frisbee, and the holding relation. In contrast, diversity-enhancing approaches such as CADS, PG, and ours ($\gamma=1.0$) struggle to preserve this relational structure. This limitation stems from the inherent trade-off where strong diversity optimization can weaken adherence to fine-grained constraints.
>
> However, a key advantage of our framework is the ability to navigate this trade-off via the attraction coefficient $\gamma$. As shown in Figure 9(e), by lowering $\gamma$ (e.g., $\gamma < 1.0$), we can intensify the anchoring force toward the initial Tweedie estimate. This effectively recovers the correct semantic alignment (the holding relation) while still offering a diversity benefit over DDIM. This demonstrates that unlike other baselines, CNO provides a controllable mechanism to resolve such failure cases when strict fidelity is prioritized. We have expanded our discussion of these limitations and their mitigation in the revised manuscript; see Section C.8 for details.

---

### Author Response · Authors · 2025-11-20
**General Response to All Reviewers**

We would like to thank the reviewers for their constructive and thorough evaluations.

We are encouraged that the reviewers highlighted **the novelty of addressing diversity at its source** by optimizing initial noise in Tweedie space (`ZwsY`, `A3GX`), **the theoretical soundness and clarity** of our mutual-information analysis and mathematical formulation (`XTmn`), **the simplicity and model-agnostic nature** of our approach (`A3GX`, `ZwsY`), and **the strong empirical performance** demonstrated across multiple T2I frameworks and text prompts (`XTmn`, `A3GX`, `ZwsY`, `qqYa`).

Below we summarize the major revisions, with detailed point-by-point responses provided for each reviewer.

---

## **Summary of Key Revisions**

- **Extended model evaluations**, including distilled and accelerated samplers such as FLUX and SDXL-Lightning (Section D.1).

- **Broader prompt coverage**, incorporating GenEval and T2I-CompBench benchmarks (Sections C.4 and D.3).

- **User study** assessing human-perceived fidelity and diversity (Section D.2).

- **Additional ablations** on key hyperparameters, including $\gamma$, batch size $B$, and downsampling rate $w$ (Sections C.2, C.4, C.5, and C.7).

- **Failure-case analysis** detailing limitations under compositionally demanding prompts (Section C.8).

- **Unified and clarified notation** for all variants of the CNO objective (throughout the manuscript).

- **Worked example** with a simplified case to facilitate understanding of our method (Section C.1).

---

### Author Response · Authors · 2025-11-28
**Reminder regarding our rebuttal and revised manuscript**

Dear Reviewers,

We would like to gently follow up to see if you have had a chance to read our response and the revised paper. We have carefully incorporated your valuable feedback, and major changes have been marked in **blue** text for your convenience.

We are eager to hear your thoughts on these revisions and are happy to answer any further questions you may have.

Sincerely, Authors

---

### Author Response · Authors · 2025-11-30
**Summary Comments to Area Chair**

**Dear Area Chair,**

We sincerely thank all reviewers for their time and thoughtful evaluations. We are encouraged that multiple reviewers found the paper **novel** (`ZwsY`, `A3GX`), **impactful** (`qqYa`), and **technically sound** (`XTmn`). In our rebuttal, we carefully **addressed all raised concerns**, providing substantial new experiments, analyses, and clarifications that directly resolve the reviewers’ feedback.

Since the discussion period was unexpectedly suspended, the reviewers did not have the opportunity to acknowledge these updates or provide follow-up feedback. Following the program chairs’ guidance that ACs should consider how reviewer impressions might have evolved, we respectfully summarize how each major concern has been resolved.

### **1. Reviewer `XTmn`: Novelty and missing baselines (e.g., TweedieMix).**
We **clarified three key contributions** of CNO. First, **CNO introduces a paradigm shift** from inference-time interventions to one-shot initial-noise optimization, enabling robust diversity enhancement even in accelerated few-step models. Second, **CNO features non-trivial extensions of InfoNCE**, including the theoretically grounded fidelity-balance parameter $\gamma$. Third, **CNO incorporates practical techniques** such as adaptive latent pooling and a stop-gradient trick, which are essential for efficient and scalable optimization. These clarifications are reflected in Sections 1 and 4.1 (highlighted in blue). Regarding the missing baselines, we **added a comparison with TweedieMix**, demonstrating that CNO achieves superior fidelity-diversity trade-offs (Section C.6).

### **2. Reviewer `qqYa`: Effectiveness on accelerated few-step models, human evaluation, and the cost of $B$**
We **expanded our evaluation to SDXL-Lightning (4-step) and FLUX-1-Schnell (4-step)**, showing that CNO performs strongly even under highly compressed few-step models, where intervention-based methods such as CADS inherently struggle (Section D.1). We also **conducted a User Study**, where CNO was consistently preferred over baselines in both alignment and diversity, aligned with the automatic metrics (Section D.2). In addition, we **examined the computational impact of batch size $B$**, demonstrating that CNO incurs only marginal overhead compared to standard samplers (Section C.7).

### **3. Reviewer `A3GX`: Clarity on the CNO loss, further ablations, user study, and failure-case analysis.**
We **provided a worked example that clarifies the CNO loss** (Section C.1), detailing the mechanisms of attraction and repulsion, the role of $\gamma$ as a fidelity controller in Tweedie space, and the gradient flow under our stop-gradient technique. We **added an additional ablation on the window size $w$** across various prompt types (Section C.4), and as noted above, we **included a User Study** showing that CNO is consistently preferred in both alignment and diversity (Section D.2). For the failure-case analysis, we examined compositional prompts and provided mitigation strategies informed by these observations (Section C.8).

### **4. Reviewer `ZwsY`: Limited evaluation domains and architectures.**
We **extended our experiments** to **GenEval** (diverse domains) and **FLUX-1-Schnell** (Rectified Flow, non-SD architecture) **(Sections D.3 and D.1)**. In both cases, CNO consistently improves diversity while preserving fidelity, supporting its generality across domains and architectures.

For convenience, we include below representative results that demonstrate the **model agnosticism** and **structural robustness** our approach, including challenging few-step regimes such as distilled and rectified flow models:

| SDXL-LT | CLIPScore ↑ | PickScore ↑ | ImReward ↑ | MSS ↓ | Vendi Score ↑ |
|---|:---:|:---:|:---:|:---:|:---:|
| **DDIM** | **31.5536** | **22.6598** | **0.7231** | 0.2865 | 2.7470 |
| **CADS** | 31.4425 | 22.5767 | 0.6876 | 0.2674 | 2.7749 |
| **CNO** | 31.4474 | 22.5659 | 0.6740 | **0.2289** | **2.8258** |

| FLUX-1 | CLIPScore ↑ | PickScore ↑ | ImReward ↑ | MSS ↓ | Vendi Score ↑ |
|:-------|:-----------:|:------------:|:-----------:|:------:|:---------------:|
| **FM-ODE** | **32.1012** | **22.7411** | **1.0499** | 0.3012 | 2.7220 |
| **CADS** | 31.7153 | 22.5431 | 0.8622 | 0.2287 | 2.8250 |
| **CNO** | 32.0664 | 22.6137 | 1.0070 | **0.2231** | **2.8316** |

Across all reviewer concerns, we have provided **direct experimental validation**, **theoretical clarification**, and **expanded analyses**. We also recognize that the unexpected freeze halted the discussion phase before reviewers could acknowledge these updates. In light of the **strong initial scores** (three “6”s and one ”4”), the **complete resolution of all concerns**, and the **demonstrated universality of CNO across diverse architectures**, we respectfully ask you to **consider these revisions** in your final assessment.

Thank you very much for your careful evaluation.

---

### Meta-Review · Area_Chair_yNck · 2026-01-08

**Summary:**

The paper addresses the limited diversity in text-to-image (T2I) diffusion models by combining Tweedie-space contrastive repulsion with anchor-guided fidelity control, referred to as Contrastive Noise Optimization.

Following the rebuttal phase, the reviewers maintained a generally positive but cautious stance, with concerns regarding: 1) novelty and comparison with missing baselines (e.g., TweedieMix), 2) the effectiveness and computational cost of CNO on accelerated few-step models, and 3) the clarity of the loss formulation alongside the need for broader evaluation across diverse architectures and human studies. Subsequently, the authors provided responses clarifying the paradigm shift to one-shot initial-noise optimization, added experimental evidence on SDXL-Lightning and FLUX-1-Schnell to prove architectural agnosticism, and included user study results as well as failure-case analyses to demonstrate the robustness and practical utility of their approach.

**Reviewer Concerns:**

The AC carefully reviewed all responses, and the authors sufficiently addressed the concerns through comprehensive experimental evidence.

**Reviewer Scores:**

Having addressed the vast majority of the reviewers' concerns, the authors' scores are expected to maintain or improve entirely positive.

---

### Decision · Program_Chairs · 2026-01-26

Accept (Poster)